



# Projections of shipping emissions and the related impact on air pollution and human health in the Nordic region

Camilla Geels[1,2], Morten Winther[1], Camilla Andersson[3], Jukka-Pekka Jalkanen[4], Jørgen Brandt[1,2], Lise M. Frohn[1,2], Ulas Im[1,2], Wing Leung[3] and Jesper H. Christensen[1,2]

[1]Department of Environmental Science, Aarhus University, Frederiksborgvej 399, P. O. Box. 358, DK-4000 Roskilde Denmark
[2] iCLIMATE, Interdisciplinary Centre for Climate Change at Aarhus University
[3]Swedish Meteorological and Hydrological Institute, SE-60176 Norrköping, Sweden
[4]Atmospheric Composition Research, Finnish Meteorological Institute, P.O. Box 503, FI-00101 Helsinki, Finland

*Correspondence to*: Camilla Geels (cag@envs.au.dk)

**Abstract.**

International initiatives have successfully brought down the emissions from shipping in Emission Control Areas (ECAs), and hence the related negative impacts on environment and human health. But the question is if this is enough to mitigate the future increase in shipping activities. The overall goal of this study is to provide an up-to-date view on future ship emissions and provide a holistic view on atmospheric pollutants and its contribution to air quality in the Nordic (and Arctic) area. First step has been to setup new and detailed scenarios for the potential developments in global shipping emissions, including different regulations and new routes in the Arctic. The scenarios include a Baseline scenario, and two additional $SO_x$ Emission Control Area (SECA) and heavy fuel oil (HFO) ban scenarios. All three scenarios are calculated in two variants involving Business As Usual (BAU) and High Growth (HiG) traffic growths. Additionally a Polar route scenario is included, with new ship traffic routes in the future Arctic with less sea ice. This has been combined with existing Current Legislation scenarios for the land-based emissions (ECLIPSE V5a) and used as input for two Nordic chemistry-transport models (DEHM and MATCH). Thereby the current (2015) and future (2030, 2050) air pollution levels and the contribution from shipping have been simulated for the Nordic and Arctic areas. Population exposure and the number of premature deaths attributable to air pollution in the Nordic area have thereafter been assessed by using the health assessment model EVA. It is estimated that within the Nordic region, approximately 9900 persons died prematurely due to air pollution in 2015. When including the projected development in both shipping and land-based emissions, this number is estimated to decrease to approximately 7900 in 2050. The shipping alone is associated with about 850 premature deaths during current day conditions (as a mean over the two models), decreasing to approximately 600 cases in the 2050 BAU scenario. Introducing a HFO ban has the potential to lower the number of cases associated with emissions from shipping to approximately 550 in 2050, while the SECA scenario has a smaller impact. The "worst case" scenario of no additional regulation of shipping emissions combined with a high growth in the shipping traffic, will on the other hand lead to a small increase in the relative impact of shipping and the number of premature deaths related to shipping is in that scenario projected to be around 900 in 2050. This scenario also leads to increased deposition of nitrogen and black carbon in the Arctic, with potential impacts on environment and climate.

## 1 Introduction

The shipping sector plays a key role for tourism and the transportation of goods in Europe and beyond (EEA, 2017). Due to the use of fossil fuels, shipping activities lead to emissions of important air pollutants like nitrogen oxides ($NO_x$), sulphur dioxide ($SO_2$), primary particles with a diameter less than 2.5 µm ($PPM_{2.5}$) and black carbon (BC). The many negative impacts related to these air pollutants and compounds subsequently formed in the atmosphere, are well established. Nitrogen deposition is a threat to sensitive ecosystems and increasing deposition is associated with loss of biodiversity (Bobbink et al., 2010), while compounds such as ozone ($O_3$), nitrogen dioxide ($NO_2$) and fine particulate matter ($PM_{2.5}$) are known to have negative impacts



on the human health (as reviewed in e.g. WHO, 2013; Pope et al., 2020), also in the Nordic area with relatively low exposure levels (Raaschou-Nielsen et al., 2020). Components like e.g. BC are also very important in relation to the climate system (Bond et al., 2013) and lead to warming in especially the Arctic region (AMAP, 2011). In recognition of the negative impacts, sulphur emissions from ships have been regulated by establishing Sulphur Emission Control Areas (SECA) for the Baltic and North

Sea as well as close to the North American coastline and Puerto Rico. In addition to these regional reductions, a global shift to low sulphur fuels was required from 2020 and onwards. This reduction was decided by the Marine Environment Protection Committee under International Maritime Organization in 2016, considering the health and climate impact of reducing sulphur emissions (Sofiev et al., 2018). Further, a NOx ECA was established for North America and new ships built after 2016 will need to comply with IMO Tier III emission requirements, which will reduce NOx emissions from these ships by 80% when

compared to Tier I level. Similar rules will be applied to the Baltic Sea and the North Sea areas from 2021 onwards. The shipping activities are nevertheless predicted to increase (e.g. Corbett et al., 2010) and the global Fourth IMO Greenhouse Gas study (Faber et al., 2020) projects a strong growth up to 2050, with GHG emission levels ranging from 90-150% of 2008 levels regardless of the measures currently in force.

According to a recent study of Sofiev et al., (2018), shipping is responsible for about 266,000 premature deaths globally, even

after the 2020 sulphur reduction is implemented. This reduction is estimated to decrease the human health effects by 137,000 deaths, especially in Asia and India, but significantly less in the northern hemisphere, where sulphur emissions are already regulated by existing SECAs. For Europe an earlier study estimated that up to 50.000 premature deaths per year can be associated with emissions from shipping (Brandt et al., 2013b). More recently a study zoomed in on the Baltic Sea region and estimated that Baltic $PM_{2.5}$ emissions from shipping caused up to 2300 premature deaths in the surrounding countries in 2016,

which was a reduction of 37% compared to before SECA was enforced in the Baltic (Barregard et al., 2019). Overall, the impacts will be largest in coastal areas and a review have previously found that shipping emissions contribute to 1-14% of the $PM_{2.5}$ levels and 7-24% of the $NO_2$ levels in coastal areas in Europe (Viana et al., 2014). Aksoyoglu et al., (2016) found the contribution from shipping to the total $PM_{2.5}$ to be largest in the western part of the Mediterranean (up to 45 %) and along the north European coast (10-15 %). However, significant contribution from ships to air quality was also reported in Madrid area,

Spain, despite the inland location of the city (Nunes et al., 2020).  In the Arctic, shipping can also be an important source for pollution in an otherwise clean and pristine environment (Schmale et al., 2018). With decreasing sea ice extend in a warming Arctic, new trans-Arctic shipping routes are becoming more likely (Corbett et al., 2010), and this can increase the traffic in the area and add to the air pollution levels in the high Arctic (Winther et al., 2014 and 2017).

In the current study we take on a Nordic perspective in order to make an updated analyses of future shipping emissions and the impacts these emissions have on health and environment. We have two overall aims:

(1) To set up shipping emission scenarios including several options to limit ship emissions, ranging from additional fuel quality requirements (Heavy fuel Oil ban), which go beyond the already agreed global sulphur cap, and to an expansion

of the existing ECA areas. Thus, the scenarios include a Baseline scenario, and two additional $SO_x$ Emission Control Area (SECA) and heavy fuel oil (HFO) ban scenarios. All three scenarios are calculated in two variants involving Business As Usual (BAU) and High Growth (HiG) traffic growths, and an additional Polar route scenario is included, with new ship traffic routes due to less Arctic sea ice in the future. The work reported here is based on vessel level modelling of ships emissions using realistic traffic data to describe the spatio-temporal variation of traffic patterns.


(2) To assess the contribution from shipping emissions to air pollution in the Nordic and Arctic area and the potential benefits of the mitigation options included in the shipping emission scenarios. This is done by applying two chemical transport models (DEHM and MATCH) set up with land-based as well as shipping emissions and analysing maps of the modelled





air pollution concentrations resulting from all emissions and the share related to shipping. It is expected that models with different physical descriptions and setups give some differences in modelled air pollution and by using two models an indication of the related uncertainties is displayed. The included emissions represent current day conditions and future projections towards 2050. For the Nordic area, we focus mainly on total $PM_{2.5}$ while for the Arctic we focus on the deposition of nitrogen and black carbon. Furthermore, the modelled concentration maps serve as input to the health impact assessment system EVA, in order to assess the overall impacts of air pollution on the human health in the Nordic area and

the changes in health impacts resulting from the different ship emission scenarios.

## 2 Materials and methods

The overall modelling setup will be described in the following. The study area and the included model domains are shown in Figure 1.

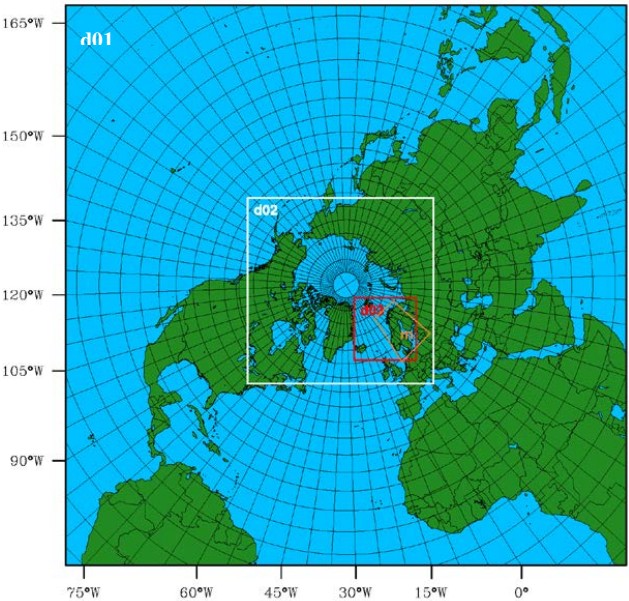

**Figure 1: The study area as defined by the three domains used in the DEHM model (d01-d03) as well as the domain used in the MATCH model (m).**

### 2.1 Setup of shipping emission inventories

The background data for the emission scenarios is, traffic data for the area north of 60N, emission factors and scenario specific emission inventories from the ship emission model developed at the Danish Centre for Environment and Energy (DCE) at

Aarhus University (Winther et al., 2017). In order to obtain a spatial coverage of the entire Nordic area and the Arctic, the DCE emission inventories are combined with a global $CO_2$ ship emission inventory produced with the Ship Traffic Emissions Assessment Model (STEAM) for 2015 (Johansson et al., 2017).

### 2.1.1 STEAM model

Global emission data from the STEAM model (developed at the Finnish Meteorological Institute) from an earlier study (Johansson et al., 2017) was applied in this work. The model uses global Automatic Identification System (AIS) data to describe shipping activity and it applies vessel level modelling using technical description of each ship in the global fleet (Jalkanen et al., 2009; 2012; Johansson et al., 2013, 2017). Details of the method and the numerical results for emissions used





in this study can be found in Johansson et al. (2017). The AIS data from terrestrial and satellite networks was purchased from
Orbcomm Ltd. (Orbcomm: 395 West Passaic Street, Suite 325 Rochelle Park, NJ 07662 USA) and the technical details
database was acquired from IHS Markit (IHS Markit Global Headquarters 4th floor Ropemaker Place25 Ropemaker Street,
London EC2Y 9LY). Emissions for ships were modelled without considering weather effects (wind, waves, ice, currents) and
thus represent ideal conditions, which may introduce uncertainties when compared to real emissions and fuel consumption.
Regardless, average absolute deviation of the STEAM predicted fuel consumption for any single ship is around 19%, whereas
inventory level totals are equivalent to those reported in the EU Monitoring, Reporting and Verification system (EU, 2015).
These comparisons were made for more than 2550 vessels reporting their fuel consumption during the reporting year 2019.

### 2.1.2 Traffic activity data in the DCE ship emission model

The ship activity data used in the DCE ship emission model are provided by the Danish Maritime Authority (DMA) based on
AIS signals received from terrestrial base stations and from satellites equipped with AIS receivers for the area north of 60N.
The data represent the years 2012-2016, divided into 0.5° longitude x 0.225° latitude grid cells, with a monthly resolution. The
ships are classified into 14 ship types and 16 ship length categories and data for total sailed distance and average sailing speeds
are provided stratified into the different ship types/length/average speed combinations that have been recorded in the individual
cells.

A weighted and consolidated ship activity data set for a base year was prepared for the DCE ship emission model based on the
five-year ship activity data provided by DMA in order to avoid inexpedient temporal and spatial specific fluctuations in traffic
records, and in order to achieve a uniform grid cell reference system for the emission projection calculations.

The traffic scaling factors used in the DCE ship emission model for traffic projections are derived from traffic growth factors
in the Corbett et al. (2010) business as usual (BAU) and high growth (HiG) scenarios by referring the DCE ship types to the
Corbett et al. (2010) ship types and by using Corbett traffic growth factors evolved from the base year. Fuel efficiency
improvements for future ships are modelled from EEDI fuel efficiency regulations agreed by Marpol 83/78 Annex VI and
mandatory from 1st January 2013 for new built ships larger than 400 GT. For further explanations regarding ship activity data,
traffic scaling factors and EEDI factors, see Winther et al. (2017).

### 2.1.3 Scenarios

The current study includes a Baseline emission projection scenario and two additional emission projections; a SECA (sulphur
emission control area) scenario and an HFO (heavy fuel oil) ban scenario (see table 1). The Baseline scenario forms the basis
for the SECA and HFO ban scenarios.

All three scenarios use the BAU and HiG growth traffic activity projections explained above in section 2.1.2 and further the
scenarios assume an increasing amount of liquefied natural gas (LNG) fuel being used as a substitution for heavy fuel oil in
the inventory area throughout the projection years. The scenarios use the "low case" LNG fuel share of total marine fuel
consumption being 2 %, 4 % and 8 % in the years 2020, 2030 and 2050, respectively, as described in the IMO 3rd GHG study
published by IMO (2015).

The Baseline scenario assumes an increase in the use of Exhaust Gas Cleaning Systems (EGCS) for $SO_2$ emission abatement
in the case of ships using HFO with a high content of sulphur. In the Baseline scenario inside the existing SECA zones (i.e.
America and North Sea/Baltic Sea SECA's) the fuel type switches from HFO to marine diesel/marine gas oil (MDO/MGO)
for ships using HFO and not having an EGCS installed (c.f. Winther et al., 2017). Outside the existing SECA's the latter ships
use 0.5 % HFO after the global sulphur cap introduction in 2020.

In the SECA scenario, the existing SECA zones (i.e. America and North Sea/Baltic Sea SECA's) are expanded to cover the
entire inventory area. The SECA scenario takes on board the Baseline shares of LNG fuel consumption and EGCS installations.



Further in the SECA scenario, the fuel is shifting from HFO to MDO/MGO outside the existing SECA's by ships using HFO and not using EGCS.

In the HFO ban scenario, no use of HFO by ships is allowed at all in the inventory area. The HFO ban scenario includes the consumption of LNG as assumed in the Baseline scenario. The remaining part of the HFO consumption not being substituted by LNG is assumed to switch to MGO/MDO in the entire inventory area.

### 2.1.4 Emission factors

The specific fuel consumption factors (SFC) and $NO_x$ emission factors (g/kWh) for HFO and MDO/MGO used in the calculations are classified according to engine type and engine production year (Ministry of Transport, 2015; MAN Energy Solutions, 2012). The $CO_2$ emission factors (g/kg fuel) comes from Nielsen et al., (2019). For LNG, the source of sfc, $NO_x$ and $CO_2$ emission factors is IMO (2015).

The $SO_2$ emission factors are proportional with the fuel sulphur content (Fs), or the sulphur removal efficiency in case of ships
with EGCS installed. For HFO fuelled ships without EGCS installed, Fs corresponds to the global IMO monitoring value of 2.45 % for 2015 (IMO, 2016) and the global fuel sulphur cap of 0.5 % for 2020 onwards. For HFO fuelled ships with EGCS installed, by assumption 2.45 % HFO is used with a removal efficiency equivalent to Fs = 0.1 %. EGCS systems are included in the emission projections from 2020 onwards (c.f. section 2.2.3). For MDO/MGO, Fs equals 0.08 % as monitored by IMO (2016).

The BC emission factors used in this project are measured values by Aakko-Saksa et al. (2016). The BC emission factors for 2.45 % HFO, 0.5 % HFO and MDO/MGO are 0.155, 0.065 and 0.056 g/kg fuel, respectively. For HFO fuelled ships with EGCS, an average BC removal efficiency of 40 % is assumed (i.e. 0.093 g/kg fuel) based on the available data from the literature (ICCT, 2015; Lack and Corbett, 2012; Johnson et al., 2016). For LNG, a BC emission factor of 0.00155 g/kg fuel is used, derived as 1 % of the BC emissions for HFO (ICCT, 2015).

The full set of basis emission factors is not shown here, but they are explained in more detail in Winther et al. (2017). However, aggregated from ship type, engine type, fuel type and engine production year, fuel related emission factors for 2015 and the forecast years 2020, 2030 and 2050 are shown in Figure 2 derived from the fuel and emission results presented above. The development of the emission factors reflects the emission technology improvements (for $NO_x$) and fuel sulphur content and the Baseline shares of LNG fuel consumption and EGCS installations (for $SO_2$ and BC). The emission factors shown in Figure
2 include the emission factor adjustments made in the calculations in order to account for engine load variations (c.f. Winther et al., 2017).

### 2.2.6 Scaling of the global $CO_2$ ship emission inventory

In order to cover the entire Nordic area and the Arctic, and make use of already well defined and elaborated scenarios for shipping in the Arctic, the global $CO_2$ ship emission inventory produced with the STEAM model for 2015 is used in
combination with the Arctic fuel consumption and emission scenarios calculated with the DCE ship emission model for 2015, 2020, 2030 and 2050 (Winther et al., 2017). The STEAM data stops at 74N, so the area beyond this is covered by the DCE model.

The ship types defined in the DCE ship emission model are mapped into the ship types defined in the STEAM model. Subsequently, the STEAM 2015 $CO_2$ emission results per ship type together with aggregated fuel related $CO_2$ emission factors
per ship type for 2015 derived from the DCE model, are used to calculate fuel consumption results specific for each ship type. A spatial distinction is made between SECA and non SECA sea areas in the calculations.

In each of the scenarios, for a given scenario year and ship type, the percentage change in total fuel consumption obtained with the DCE model from 2015 to the scenario year, is then used to scale the 2015 STEAM model fuel consumption, in order to calculate STEAM related fuel consumption results for the scenario year in question.





STEAM related emission results are subsequently obtained as the product of 1) the aggregated emission factors obtained with
the DCE ship emission model for each scenario, scenario year and STEAM ship type and 2) the corresponding fuel
consumption. The final emission data set for 2015 and the scenario years are monthly files with a spatial resolution of 0.1°×0.1°.

### 2.3 The DEHM model

The Danish Eulerian Hemispheric Model (DEHM) is a state-of-the-art three-dimensional, Eulerian, atmospheric chemistry
transport model (CTM) originally developed in the early 1990's to study the atmospheric transport of Sulphur-dioxide and
Sulfate into the Arctic (Christensen, 1997). The model has been modified, extended and updated continuously since then and
now includes a comprehensive chemical scheme, detailing 80 chemical species and 158 chemical reactions including a Volatile
Basis Set (VBS) for describing Secondary Organic Aerosols (SOA) (see e.g. Brandt et al., 2012 for a detailed description of
DEHM and Bergstrøm et al., 2012 for a description of the VBS part). In the current study DEHM has been setup with a main
domain covering the Northern hemisphere and part of the Southern hemisphere, with a resolution of 150 km x 150 km. Within
this domain a nested domain (d02) with a resolution of 50 km x 50 km is covering the Arctic and Europe and finally a second
nest (d03) with a resolution of 16.67 km x 16.67 km is covering the Nordic region (see Figure 1).
The meteorological data, driving DEHM, is calculated using the WRF model (Skamarock et al., 2008), set up with the same
domains and nests as DEHM and forced with ERA-Interim meteorology (Dee et al., 2011). Natural emissions like sea salt and
biogenic volatile organic compounds (VOCs) are calculated online in the model as a function of meteorological parameters
(described in Soares et al., 2016, Zare et al., 2012 and Zare et al., 2014). The anthropogenic emissions for the current and
future periods are based on the global 0.5° x 0.5° ECLIPSE V5a data sets including sectoral emissions
(https://www.iiasa.ac.at/web/home/research/researchPrograms/air/Global_emissions.html). We apply the future Baseline
scenario assuming Current LEgislation (CLE) for the air pollution components (see Klimont et al., 2017 for an overview). The
projected change in the land-based emissions in the applied domains in DEHM, are given in table S1 in the Supplement to this
paper. The shipping emissions in ECLIPSE V5a have been replaced with the new shipping emissions described in Section 2.1.
We have run the model for the meteorological year 2015 (and December 2014 as spin up) with a combination of land based
ECLIPSE V5a and new shipping emissions representing the years 2015, 2030 and 2050. This is done in order to isolate the
impact from emission changes. Additionally, we have made simulations, with and without a new polar diversion route and
simulations where the shipping emissions have been reduced by 30% (i.e. multiplied by 0.7). By scaling the results afterwards,
the impact from shipping alone can be analyzed, but non-linear effects of atmospheric chemistry are still included. In total 22
simulations have been made with the DEHM model (an overview of model runs is given in Table S2 in the Supplement). The
Baseline simulation with 2015 BAU emissions has been evaluated by comparison to European observations of the components
relevant for the health assessment ($PM_{2.5}$, $NO_2$, and $O_3$) and a sufficiently good agreement between model and observations is
seen both in terms of level and variability (see the Supplement for details). Furthermore, the DEHM model is one of the core
models in the Copernicus Atmosphere Monitoring Service (CAMS) providing daily air pollution forecasts and analyses, which
are continuously evaluated on-line against European observations (see https://www.regional.atmosphere.copernicus.eu/).

### 2.4 The MATCH model

The MATCH (Multi-scale Atmospheric Transport and CHemistry) model (Robertson et al., 1999; Andersson et al., 2007;
2015) is a state-of-the-art Eulerian chemistry and transport model, including wet-, thermal- and photochemical reactions, to
describe the sulfur and nitrogen cycle as well as tropospheric ozone chemistry, and particle formation and transformation. The
version used in the current set up is the same as was used in the EURODELTA-trends exercise (Colette et al., 2017). It includes
online emissions of sea salt as described in Soares et al. (2016). Secondary organic aerosol formation is modelled through a
volatility basis set and biogenic VOC emissions are modelled online, both as described by Simpson et al. (2012). Further
details on the model configuration are described in Colette et al. (2017). The driving meteorological forcing data used was the





HIRLAM operational weather data for 2015 for the domain covering Fennoscandia (11 km resolution). The lateral and top boundaries of this domain were fed every 6 hours by results from the DEHM model, domain d01 (see Figure 1). The anthropogenic and shipping emissions were the same as for DEHM, but with MATCH we simulate the current (2015) and a selection of the 2050 shipping scenarios. The MATCH grid is smaller than the DEHM d03 grid with a finer horizontal resolution (11 km x 11 km). The shipping attribution was conducted in the same manner as with the DEHM model, by reducing the shipping emissions by 30% and using the corresponding DEHM simulation on the boundary of the MATCH grid. In total 12 simulations have been made with the MATCH model (Table S2 in the Supplement).


The current model configuration is extensively evaluated with in situ observations and compared to the performance of other models in numerous papers in similar set ups as in this study for ozone, particles, and nitrogen and sulfur deposition in Europe (Otero et al., 2018; Theobald et al., 2019; Vivanco et al., 2018, Ciarelli et al., 2019a,b). The conclusion is that MATCH in the current configuration, performs among the best models for near-surface ozone as well as nitrogen and sulfur deposition, and for particles, it displays high correlation with observations while PM concentrations are somewhat underestimated. An evaluation of MATCH for the Fennoscandia region for the year 2015 is included in the Supplement of this paper. MATCH is also a core model in the operational CAMS (Marécal et al., 2015; https://www.regional.atmosphere.copernicus.eu/) including daily updates of daily air pollution forecasts, fused measurement and modelling of atmospheric concentrations (through data assimilation), as well as model performance scores. MATCH is a building block in the MATCH Sweden system for environmental surveillance (Andersson et al., 2017), including measurement model fusion of total atmospheric deposition (MMF-TDEP) of ozone, nitrogen, sulphur and base cations.



### 2.5 The EVA system


The EVA (Economic Valuation of Air pollution; Brandt et al., 2013a;b; Geels et al., 2015, Im et al., 2018) model system is based on the impact pathway chain (Friedrich and Bickel, 2001), where modelled air pollution levels are coupled to population data for calculation of human exposure, health impacts (both mortality and morbidity), and related external costs. In the current study, we focus on the health impacts and are not including the assessment of the cost. The health impacts are calculated using linear exposure-response functions, which in the applied model version (EVAv5.2) are based on the HRAPIE recommendations (WHO, 2013). The number of premature deaths in the system is calculated from short-term exposure to $O_3$, $NO_2$, $SO_2$ and $PM_{2.5}$ (acute deaths) as well as long-term exposure to $PM_{2.5}$ and $NO_2$ (chronic deaths). The EVA model system can be used to estimate the health impacts due to the total air pollution levels or due to contributions from various model scenario runs, where e.g. specific emission sectors are reduced, or the weather/climate scenarios are changed. In this study, the population exposure has been assessed for the d03 DEHM domain by combining the concentration maps from both DEHM and MATCH with gridded population data from EUROSTAT for 2015 (see the Supplement for a table (S4) with the national total populations and a map of the population distribution in the applied 16.67 km x 16.67 km grid). In order to do so, the MATCH 11 km x 11 km gridded data has been aggregated to the d03 grid. The EVA model system has been compared to other similar models (Anenberg et al., 2015; Lehtomäki et al., 2020), is part of the Danish monitoring programme (Ellermann et al., 2020) and has been used routinely in numerous advisory projects for public authorities. In Section 3.2 the current results for 2015 are compared with EEA's results for the same year (EEA, 2018).




### 3 Results

In this section we first describe the projected developments in the shipping emissions in the different scenarios within the Nordic area. Thereafter we describe the modelled current day air pollution levels with focus on total $PM_{2.5}$ and the overall health impact related to air pollution in the Nordic area. We then move on to the future developments in the $PM_{2.5}$ levels as simulated by the two models based on the shipping scenarios as well as the projected development in the land-based emissions.




The related impacts on the number of premature deaths are then analyzed. Next we focus more directly on the contributions from shipping in terms of the PM$_{2.5}$ levels and health impacts. Finally we move to the Arctic and demonstrate, how the deposition of BC and Nitrogen will be effected by the projected developments in emissions.

### 3.1 Development in shipping emissions

Table 1 shows the shipping emissions for the domain area d03 for the base year 2015, and the Baseline, SECA and HFO ban BAU scenarios in 2020, 2030 and 2050. The percent changes between Baseline and SECA/HFO ban scenario results in 2020, 2030 and 2050 are also shown in Table 1.

**Table 1: Total shipping emissions for the Nordic (d03) domain for the base year 2015, and the Baseline, SECA and HFO ban BAU scenarios in 2020, 2030 and 2050, and percent changes between Baseline and SECA/HFO ban scenario results in 2020, 2030 and**
**2050.**

| | | | BAU | | | | | HiG | | | | |
|---|---|---|---|---|---|---|---|---|---|---|---|---|
| | Scenario | Year | Fuel | CO$_2$ | SO$_2$ | NO$_x$ | BC | Fuel | CO$_2$ | SO$_2$ | NO$_x$ | BC |
| | | | M tonnes | M tonnes | K tonnes | K tonnes | K tonnes | M tonnes | M tonnes | K tonnes | K tonnes | K tonnes |
| Total | Base year | 2015 | 9.6 | 30.3 | 59.6 | 719 | 0.80 | 9.6 | 30.3 | 59.6 | 719 | 0.80 |
| | Baseline | 2020 | 10.2 | 32.1 | 23.4 | 739 | 0.75 | 10.7 | 33.9 | 24.7 | 781 | 0.79 |
| | | 2030 | 10.9 | 34.3 | 24.1 | 574 | 0.81 | 12.5 | 39.2 | 27.5 | 661 | 0.93 |
| | | 2050 | 13.3 | 41.6 | 27.2 | 292 | 0.97 | 18.3 | 57.0 | 37.5 | 400 | 1.35 |
| % change | SECA | 2020 | 0 | 0 | -31 | 0 | -2 | 0 | 0 | -31 | 0 | -2 |
| | | 2030 | 0 | 0 | -29 | 0 | -2 | 0 | 0 | -29 | 0 | -2 |
| | | 2050 | 0 | 0 | -27 | 0 | -1 | 0 | 0 | -27 | 0 | -1 |
| | HFO ban | 2020 | 0 | 0 | -33 | 0 | -9 | 0 | 0 | -33 | 0 | -9 |
| | | 2030 | 0 | 0 | -32 | 0 | -12 | 0 | 0 | -32 | 0 | -13 |
| | | 2050 | 0 | 0 | -31 | 0 | -16 | 0 | 0 | -31 | 0 | -17 |

### 3.1.1 Baseline results

The emission development relies on the development in fuel consumption and the corresponding emission factors. Figure 3 shows the emissions of CO$_2$, NO$_x$, SO$_2$ and BC for 2015 and Baseline scenario results for 2020, 2030 and 2050, split into the
current ECA and non ECA parts of domain d03.

In terms of total fuel consumption, the projected growth in ship traffic during the forecast period more than outweighs the future fuel efficiency improvements for ships. From 2015 to 2050, the fuel consumption changes for Total *[SECA, non SECA]* becomes 39% *[43%, 20%]* for BAU traffic growth and 91% *[98%, 51%]* for HiG traffic growth, respectively. Almost identical
percentage changes are calculated for CO$_2$ emissions from 2015 to 2050 due to the almost constant fuel dependent emission factors for CO$_2$ in the projection period (Figure 2).

For NO$_x$ the Total *[NECA, non NECA]* emission changes from 2015 to 2050 are -59% *[-70%, 1%]* for BAU traffic growth and -44% *[-58%, 30%]* for HiG traffic growth, respectively. The total NO$_x$ emission reductions during the forecast period are due to the decrease in NO$_x$ emission factors (Figure 2). NO$_x$ emission reductions are most significant for the existing NECA area,
where new engines installed on board ships from 1 January 2021 must comply with the most stringent IMO Tier III NO$_x$ emission standards.

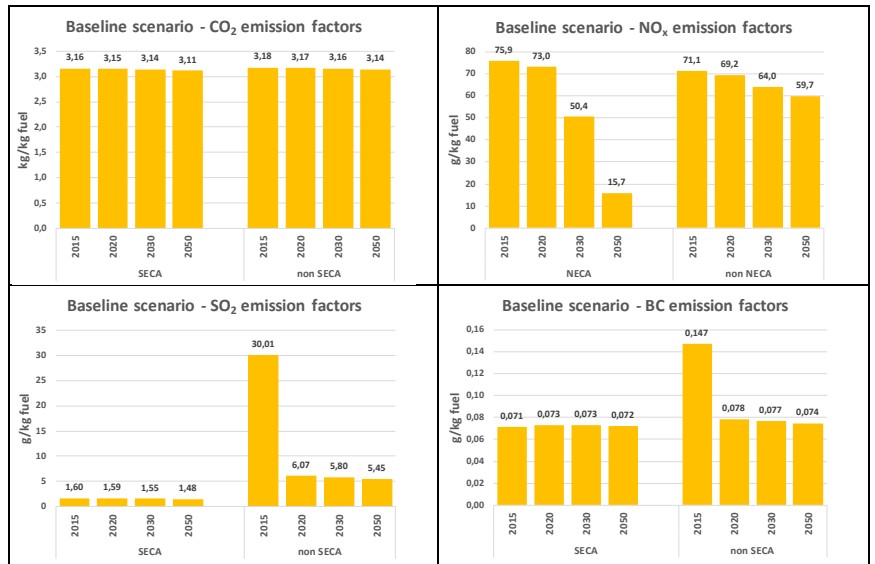

**Figure 2: Fuel related emission factors of $CO_2$, $NO_x$, $SO_2$ and BC for 2015, 2020, 2030 and 2050 split into SECA and non SECA.**

The Total *[SECA, non SECA]* $SO_2$ emission changes from 2015 to 2050 are -54% *[32%, -78%]* for BAU traffic growth and -37% *[86%, -71%]* for HiG traffic growth, respectively. For BC, the Total *[SECA, non SECA]* emission changes from 2015 to 2050 become 21% *[45%, -39%]* for BAU traffic growth and 69% *[105%, -22%]* for HiG traffic growth, respectively.

For $SO_2$ and BC, the major reason for the emission reductions outside SECA from 2015 to 2050 is the shift from HFO with a Sulphur content of 2.45% in 2015 to HFO with 0.5% Sulphur from 2020 onwards and the consequently reduced emission
factors.

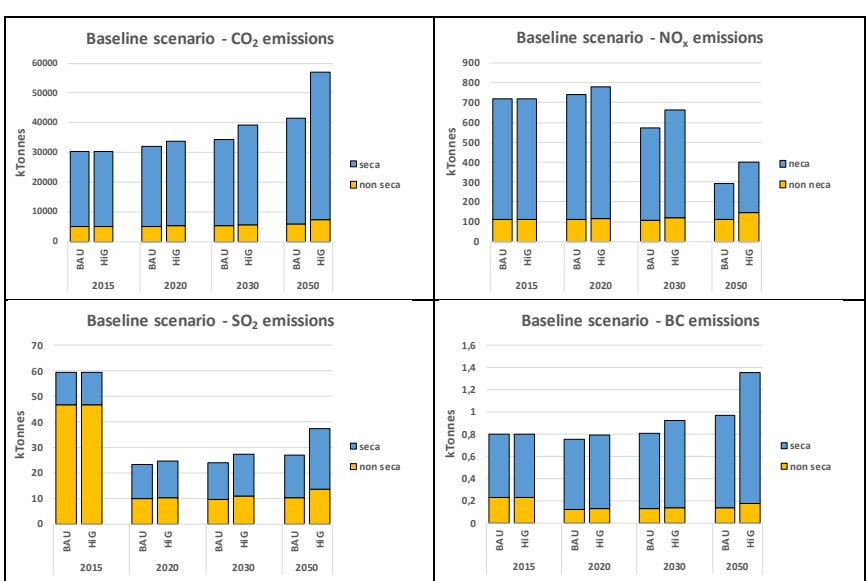

**Figure 3: $CO_2$, $NO_x$, $SO_2$ and BC emissions for 2015 and Baseline scenario results for 2020, 2030 and 2050, split into SECA and non SECA parts of domain d03.**

Inside SECA, the emissions of BC increase somewhat more than what is expected from the changes in fuel consumption due to higher BC emission factors for HFO in combination with EGCS compared with the emission factors for the MDO/MGO





fuel being replaced. The $SO_2$ emissions increase slightly less than fuel consumption due to the gradually increased

consumption of liquefied natural gas (LNG) in the Baseline scenario.

**3.1.2 Emission and fuel consumption results across scenarios**

The fuel consumption and $NO_x$ emission totals calculated for the SECA and HFO ban scenario equal the results obtained in

the Baseline scenario (Table 1; Figure 4 ($NO_x$)). The main reason for this is that the engine specific fuel consumption and $NO_x$

emission factors are unaffected by the fuel switch from HFO to MDO/MDO, and also the same shares of LNG of total fuel

consumption per forecast year is assumed in both scenarios (c.f. scenario definitions). Very small $CO_2$ emission differences

between the Baseline, SECA and HFO ban scenarios are the results of small differences in the fuel related $CO_2$ emission factors

for HFO and MDO/MGO.

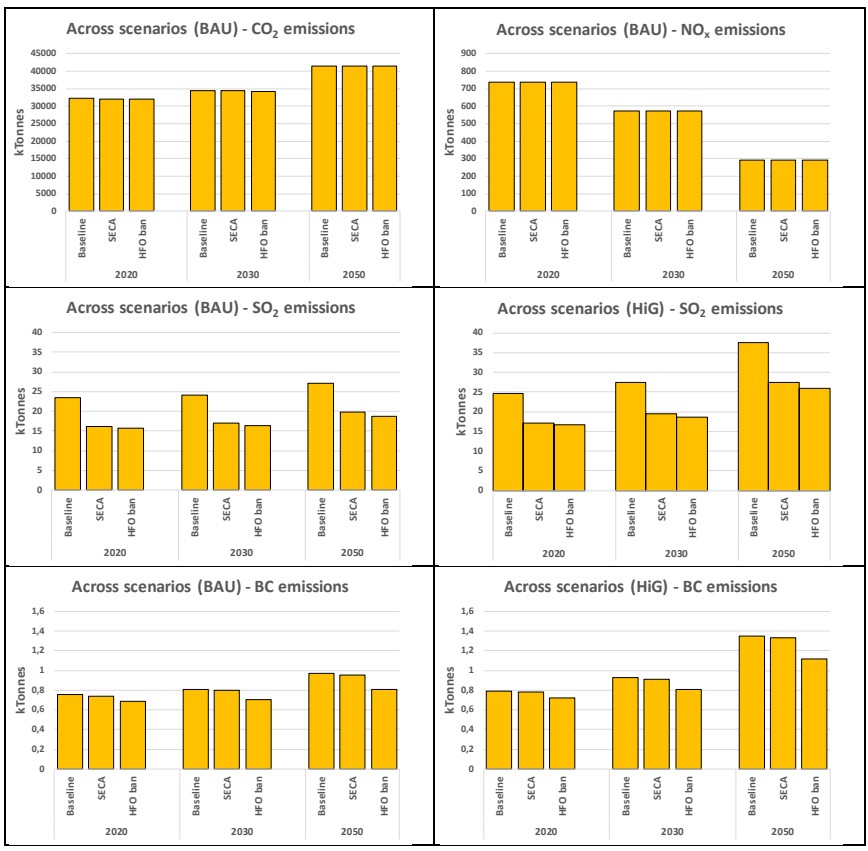


**Figure 4: Emissions for domain d03 calculated in the Baseline, SECA and HFO ban BAU scenarios ($CO_2$, $NO_x$, $SO_2$ and BC) and the HiG scenarios ($SO_2$ and BC).**

In all scenario years for $SO_2$, the calculated emissions for the SECA and HFO ban scenarios are close to 30% lower than the

emissions calculated for the Baseline scenario (Table 1). In the SECA scenario, HFO is only used by ships with EGCS, with

a Sulphur removal efficiency equivalent Fs = 0.1 (Section 2.1.3). The 0.5% fuel oil used outside the original SECA area, which

is not being replaced by LNG, is replaced by MDO/MGO (Fs = 0.08) according to the scenario definitions. In the HFO ban

scenario, all HFO consumption by ships not being replaced by LNG is replaced by MGO/MDO.





For BC in 2020*[2030, 2050]*, the SECA scenario emissions are 2%*[2%, 1%]* lower than the Baseline results (Table 1) in both

traffic growth cases. For the HFO ban scenario in 2020*[2030, 2050]* with BAU traffic growth scenario, the BC emissions are

9%*[12%, 16%]* smaller than the Baseline results (Table 1). For HiG traffic growth, the BC emissions become 9%*[13%, 17%]*

smaller than the Baseline results.

Apart from LNG with similar fuel consumption shares assumed in all three scenarios, in the HFO ban scenario, only the fuel

type MDO/MGO with the smallest BC emission factor (0.053 g/kg fuel, before load adjustment) is used. In the Baseline and

SECA scenarios similar shares of EGCS are used with a higher BC emission factor (0.093 g/kg fuel, before load adjustment).

In the SECA scenario, however, the BC emissions from MDO/MGO fuel that replaces 0.5% fuel oil are smaller due to the

level of the BC emission factors.

### 3.1.3 Results for diversion routes

Potential changes in polar sea ice distribution and quantity due to a future warming, can open up for additional ship traffic in

the Arctic diverted from current shipping routes. Therefore scenario estimates of $CO_2$ (proxy for fuel consumption), BC, $NO_x$

and $SO_2$ for so called 'diversion traffic' are made based on the Business As Usual (BAU) and High Growth (HiG) scenario

emission results for the diversion routes[1] from Corbett et al. (2010). Based on the $CO_2$ emissions for the diversion routes and

the fuel related emission factor for $CO_2$ from Corbett et al. (2010), the diversion route fuel consumption is calculated. Fuel

consumptions are then further modified by taking into account future fuel efficiency improvements for ships.

Next, the diversion route emissions related to the Baseline, SECA and HFO ban scenarios for BAU traffic and the Baseline

scenario for HiG traffic, respectively, are calculated as the product of the diversion route fuel consumption and the emission

factors derived from each of the four scenarios.

Table 2 shows the estimated emissions of $CO_2$, $SO_2$, $NO_x$ and BC for the polar diversion routes passing through domain d03

in 2020, 2030 and 2050 for the Baseline, SECA and HFO ban emission scenarios based on the BAU traffic growth scenario

and the Baseline emission scenario based on the HiG traffic growth scenario, respectively. The diversion emission results are

also shown in Figure 5 together with the totals without including the diversion emission contribution (Table 1) for each scenario

case.

**Table 2: Estimated emissions of $CO_2$, $SO_2$, $NO_x$ and BC for the polar diversion routes in domain d03 in 2020, 2030 and 2050.**

| Scenario | Year | Diversion emission contribution | | | | % emission increase due to diversion | | | |
|---|---|---|---|---|---|---|---|---|---|
| | | $CO_2$ | $SO_2$ | $NO_x$ | BC | $CO_2$ | $SO_2$ | $NO_x$ | BC |
| | | M tonnes | K tonnes | K tonnes | tonnes | % | % | % | % |
| BAU Baseline | 2020 | 0.92 | 1.41 | 24.7 | 23 | 3 | 6 | 3 | 3 |
| | 2030 | 1.06 | 1.53 | 22.3 | 26 | 3 | 6 | 4 | 3 |
| | 2050 | 3.67 | 4.78 | 48.1 | 87 | 9 | 18 | 16 | 9 |
| BAU SECA | 2020 | 0.91 | 0.46 | 24.7 | 21 | 3 | 3 | 3 | 3 |
| | 2030 | 1.06 | 0.52 | 22.3 | 24 | 3 | 3 | 4 | 3 |
| | 2050 | 3.66 | 1.70 | 48.1 | 81 | 9 | 9 | 16 | 9 |
| BAU HFO ban | 2020 | 0.91 | 0.45 | 24.7 | 19 | 3 | 3 | 3 | 3 |
| | 2030 | 1.05 | 0.50 | 22.3 | 22 | 3 | 3 | 4 | 3 |
| | 2050 | 3.65 | 1.61 | 48.1 | 69 | 9 | 9 | 16 | 9 |
| HiG HFO Baseline | 2020 | 1.19 | 1.81 | 32.0 | 30 | 4 | 7 | 4 | 4 |
| | 2030 | 3.06 | 4.40 | 64.6 | 76 | 8 | 16 | 10 | 8 |
| | 2050 | 18.77 | 24.53 | 246.4 | 451 | 33 | 65 | 62 | 33 |

---

[1] In Corbett et al. (2010), BAU diversion traffic are 1%, 1% and 1.8% of global shipping in the forecast years 2020, 2030 and 2050,

respectively. HiG growth diversion traffic are 1%, 2% and 5% of global shipping in the forecast years 2020, 2030 and 2050, respectively.






Based on the BAU diversion traffic scenarios for the forecast years 2020 *[2030, 2050]*, the additional percentage of emissions from ship traffic following the Arctic diversion routes in all three scenarios is +3% *[+3%, +9%]* for $CO_2$ and BC, +3% *[+4%, +16%]* for $NO_x$, +6% *[+6%, +18%]* for $SO_2$ (Baseline) and +3% *[+3%, +9%]* for $SO_2$ (SECA and HFO ban).

The additional percentage of emissions from ship traffic on Arctic diversion routes based on HiG diversion traffic for the
forecast years 2020 *[2030, 2050]* becomes +4 % *[+8 %, +33 %]* for $CO_2$, +4 % *[+8 %, +33 %]* for BC, +7 % *[+16 %, +65 %]* for $SO_2$ and +4 % *[+10 %, +62 %]* for $NO_x$.

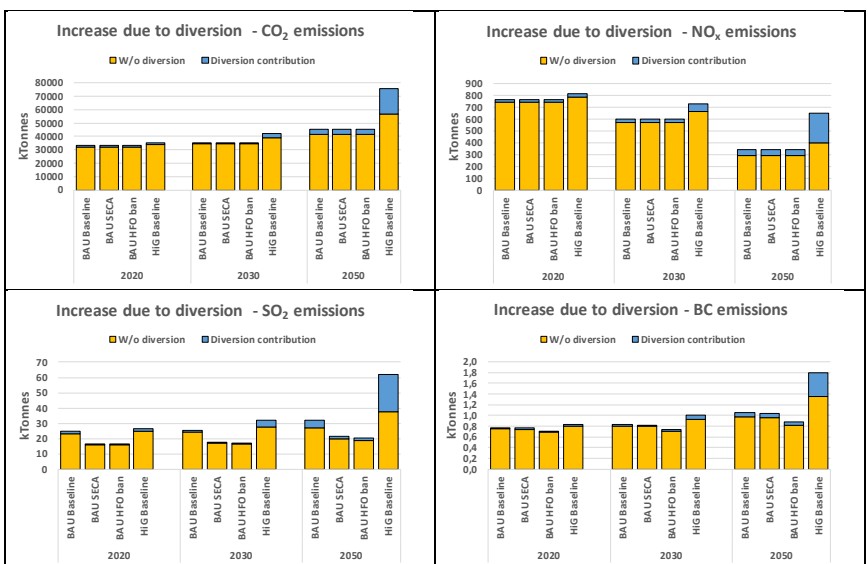

**Figure 5: Estimated emissions of $CO_2$, $SO_2$, $NO_x$ and BC in domain d03 without polar diversion traffic and for polar diversion traffic for the scenarios in 2020, 2030 and 2050.**

### 3.2 Current day Nordic air pollution and health effects


The annual mean $PM_{2.5}$ concentration as modelled with the two models in the Baseline current situation (2015) is shown in Figure 6a and 6b. The same overall pattern is seen in the two maps with highest $PM_{2.5}$ levels in the southern part of the domain and lower values towards North. Thus, land-based emissions and long-range transport are major contributing factors for the overall $PM_{2.5}$ levels and hence for human exposure in the Nordic region in 2015. The concentration levels are highest in the
DEHM results for 2015, while the MATCH results are somewhat lower than DEHM.

The total number of premature deaths attributable to short term exposure to $O_3$, $NO_2$, $SO_2$ and $PM_{2.5}$ (acute effects) as well as long-term exposure to $PM_{2.5}$ (chronic effects) for the base year 2015 are shown in Figure 7 for Norway, Finland, Denmark and Sweden. Concentration fields from both the DEHM and the MATCH model have been used as input to the EVA model and
the results are given as box plots in order to show the central (mean) estimate as well as the range between the two models. The upper estimate is for all countries obtained by the DEHM-EVA setup, while the MATCH-EVA setup gives a slightly lower estimate. The total number of premature deaths in the Nordic region in 2015 is estimated to be between ca. 9400 (MATCH-EVA) and ca. 10.400 (DEHM-EVA) (or 9900 ± 10% - given as the average of the two models and the difference as percent). The number is lowest for Norway (1300±8%) and Finland (1700±2%), while highest for Sweden (3600±7%).



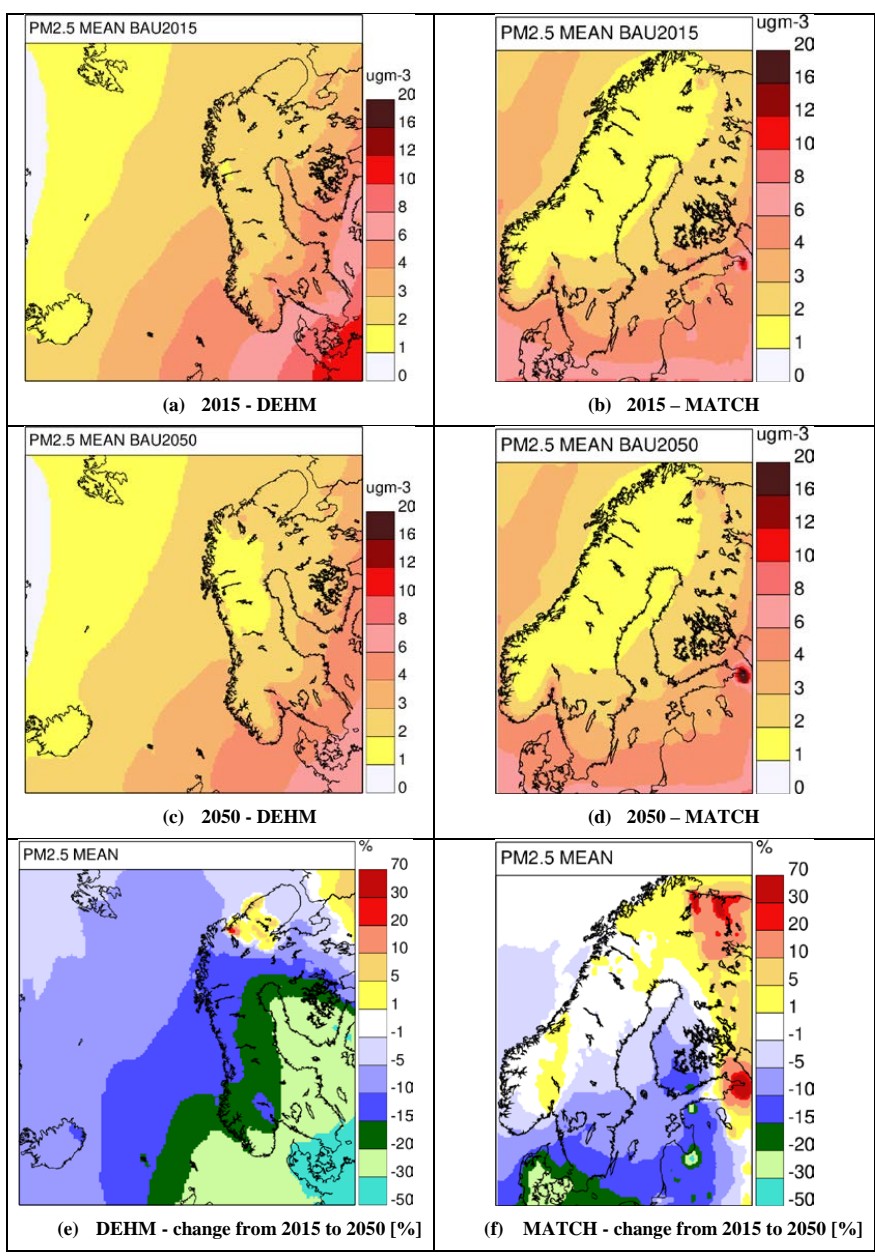

**Figure 6: The annual mean surface PM$_{2.5}$ concentration [µg/m$^3$] for 2015, 2050 and the % change as simulated by the two models.**

For Denmark the total number of premature deaths is only slightly lower (3300±3%). Part of the difference between the countries is of course related to difference in the population numbers, where Sweden with a population of about 9.9 million (in 2015) by far is the largest in the Nordic region (see table S4). The high number of premature deaths in Denmark, where the total population is only slightly higher than in Norway, can be explained by the higher air pollution levels across Denmark (see Figure 6a and 6b). In all countries the number of premature deaths attributable to long-term exposure to PM$_{2.5}$ is a factor of 2-4 higher than the number of premature deaths attributable to acute effects. The DEHM-EVA setup also covers Iceland



and the Faroe Islands, where the total number premature deaths are estimated to be approximately 40 and approximately 15,
respectively, in 2015. MATCH-EVA excludes these regions, to the benefit of a higher resolution in a smaller domain.

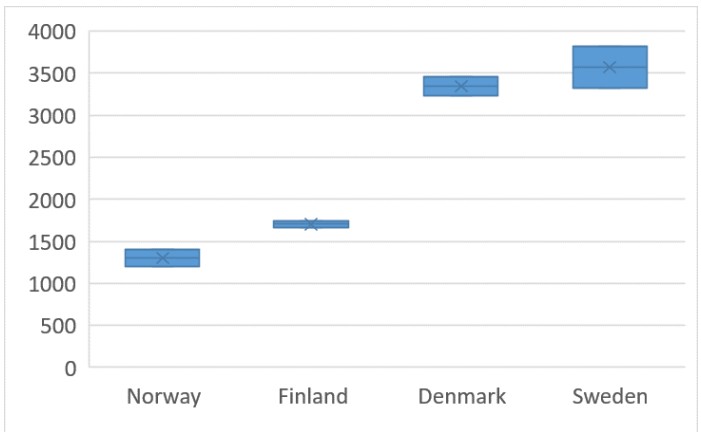

**Figure 7: Premature deaths attributable to NO$_2$, O$_3$, SO$_2$ and PM$_{2.5}$ exposure in the four largest Nordic countries in the base case for 2015. The results are given as box plots, where the 'x' indicates the central estimate and the size of the box show the difference**
**between the DEHM-EVA and MATCH-EVA assessments.**

The current assessment can be compared to the recent European Environment Agency (EEA) "Air quality in Europe — 2018
report" (EEA, 2018), where premature deaths attributable to PM$_{2.5}$, NO$_2$ and O$_3$ exposure have been estimated for 41 European
countries for the year 2015. The EEA specifies that the uncertainty related to the health estimates is ±35 % (PM$_{2.5}$), ±45 %
(NO$_2$) and ±50 % (O$_3$). In Figure 8 the EEA estimates for PM$_{2.5}$ mortality in four of the Nordic countries are compared to the
current DEHM-EVA and MATCH-EVA estimates based on the Baseline 2015 simulations. Overall, the estimates are very
similar for the four countries and some differences should be expected due to methodological differences.

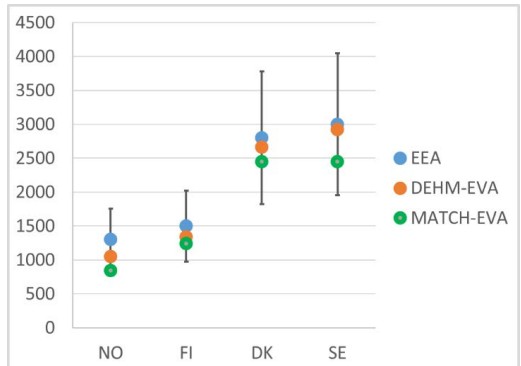

**Figure 8: Comparison between the assessment of premature deaths reported by EEA for 2015 and the current assessment (base**
**case 2015) for effects related to PM$_{2.5}$ exposure. The EEA number includes an error bar displaying the assumed uncertainty**
**(±35%).**

### 3.3 Future developments in Nordic air pollution and health effects

The annual mean PM$_{2.5}$ concentration as simulated by both models for the BAU Baseline 2050 scenario is seen in Figure 6c
and 6d. The land-based emissions of e.g. NO$_x$ and SO$_x$ are projected to decrease in most of Europe in the applied ECLIPSE
V5a scenarios (see Table S1 in the Supplement). This leads to significant general reductions in the annual mean PM$_{2.5}$ levels
in most parts of the Nordic region towards 2050 (see Figure 6e and 6f for the change in %). Parts of Russia (e.g. around





Murmansk and St Petersburg) and parts of Norway are standing out as exceptions, where the emissions and hence the PM$_{2.5}$ concentration is projected to increase. In the MATCH results, the concentration in the Oslo region is projected to increase by a few percent. This is not seen in the DEHM results. Potential causes to this are the slightly lower resolution in the DEHM

setup, differences in chemical scheme and differences in long-range transported (LRT) component from continental Europe (the LRT-component is weaker in MATCH), possibly partly due to differences in meteorological forcing. Thus, although there is a general decrease in exposure to PM$_{2.5}$, some areas are projected to possibly experience increased exposure.

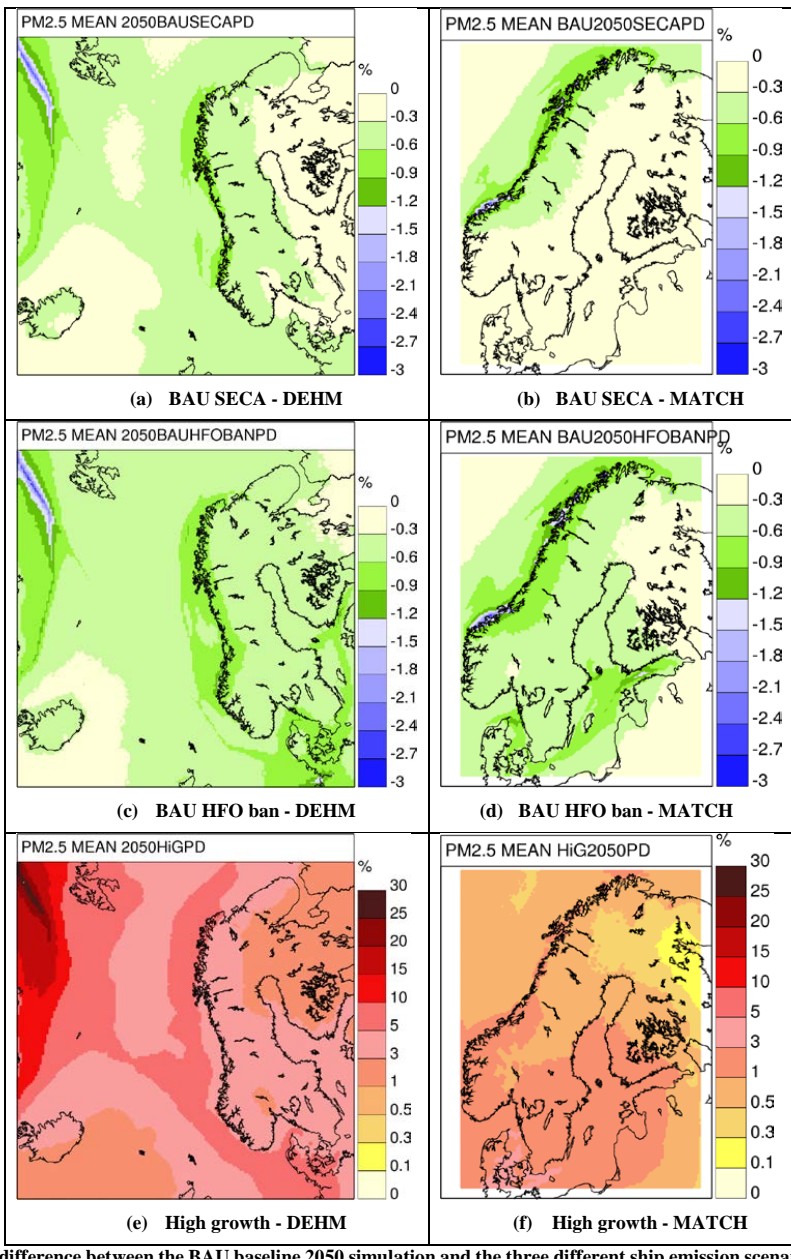

**Figure 9: The % difference between the BAU baseline 2050 simulation and the three different ship emission scenarios (including the polar diversion route). As simulated by the two models DEHM (left) and MATCH (right). Calculated as e.g. (2050BAU_SECA -**

**2050BAU)/2050BAU * 100%.**



For 2050 the difference between the BAU Baseline simulations and the simulations including the BAU SECA, BAU HFO ban and HiG Baseline shipping scenarios (land based emissions are unchanged in all scenarios) can be analysed in detailed based
on the %-difference maps given in Figure 9.

With the SECA scenario (9a and 9b), a small decrease in the $PM_{2.5}$ concentration is seen outside the current Baltic/North Sea SECA areas compared to the BAU scenario. Along the Norwegian coast the two models project a decrease in $PM_{2.5}$ concentration between 0.6-1.5%, whereas smaller changes are seen across the other countries.

In the HFO ban scenario (9c and 9d) a slightly larger decrease is seen in most of the domain and the $PM_{2.5}$ levels along shipping
routes in the Baltic and around Denmark are up to 1.5% lower than in the Baseline, and like in the SECA scenario the largest decrease (ca 2%) is seen along part of the Norwegian coast in the MATCH simulation. These changes in the $PM_{2.5}$ levels are as described in Section 3.1.2 linked to decreased $SO_2$ emission in both the SECA and HFO ban shipping scenario, which leads to a decrease in the formation of secondary inorganic aerosols and hence in the total PM mass. The HFO ban scenario also includes a decrease in the BC, which leads to small decrease in the primary PM part in the models.

The HiG scenario will on the other hand, result in an increase in the $PM_{2.5}$ concentration compared to the BAU scenario (Figure 9e and 9f). Here an increase in the $PM_{2.5}$ levels of up to 1-5% are seen across Finland, Norway and Sweden and even higher over Denmark, according to the DEHM results. Somewhat lower values are seen in the MATCH results.

The resulting development in the number of premature deaths in the Nordic region is seen in Figure 10a, where the total number for the Nordic region in the current Baseline for 2015 is shown together with the total number in the BAU scenarios
for 2030 (only DEHM) and 2050 (DEHM and MATCH). The total number of 9900 ± 10% in 2015 is hence projected to decrease to about 8300 in 2030 and further to 7900± 6% in 2050 in the current study. The MATCH-EVA setup gives the highest number of premature deaths in 2050 (8200 premature deaths), mainly due to larger areas of predicted increases in $PM_{2.5}$ than DEHM (7700 premature deaths).

In Figure 10b the resulting differences in the number of premature deaths in 2050 is shown. Only a small decrease (>-20
premature deaths) is seen for the SECA scenario. The HFO ban scenario has a somewhat larger effect and would decrease the number of premature deaths in 2050 with almost 50 cases in the Nordic area compared to the BAU Baseline. The HiG scenario on the other hand, is projected to increase the number of premature deaths in the Nordic area by approximately 300 cases in 2050.

**3.4 The contributions from shipping**

The specific model simulations with shipping emissions reduced by 30%, can for each scenario be used to quantify the contributions from shipping to both the air pollution levels and the related health impacts. In terms of the annual $PM_{2.5}$ concentration, the contribution from shipping for all scenarios (given as % of total $PM_{2.5}$ in Figure 11) is highest along shipping routes and in coastal regions giving a similar spatial pattern for both the DEHM and the MATCH models. For the current 2015 Baseline about 7-15% of the annual mean $PM_{2.5}$ concentration over the majority of the Nordic land areas is linked to shipping
in the DEHM results, while the MATCH results point to 1-10% for the same areas. This is in line with a recent study based on the EMEP model, where the contribution from shipping to the averaged $PM_{2.5}$ concentration in the Nordic countries ranged from about 5% in Finland to about 13% in Denmark (Jonson et al., 2020).

From Figure 11c-11g it is seen that only small changes are projected for this contribution in the future according to the MATCH model, while the DEHM results for both the BAU 2050 scenario and the HFO ban 2050 scenario show a decrease in part of
the area. The increased traffic in the HiG Baseline scenario increases the overall contribution from shipping, so it is higher than in 2015.



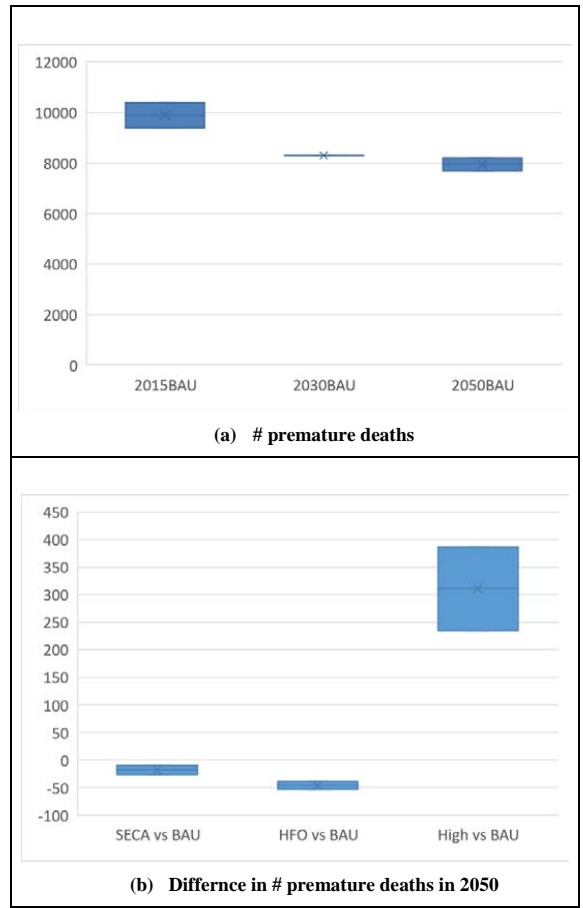

(a) # premature deaths

(b) Differnce in # premature deaths in 2050

**Figure 10: (a) The estimated total number of premature deaths due to air pollution (sum for the largest four Nordic countries) for 2015 and the projected numbers for 2030 (based on DEHM-EVA alone) and 2050. (b) The difference in the number of premature deaths between the ship emission scenarios for 2050.**

The estimated number of premature deaths attributable to shipping can be seen in Figure 12. As a mean over the two models

this amounts to approximately 850 premature deaths under current day conditions, decreasing to just below 600 cases in the 2050 BAU Baseline scenario and to about 580 and 550 in the BAU SECA and BAU HFO ban scenarios. In the HiG Baseline scenario this number is projected to increase to almost 900 cases, and hence to a value slightly higher than today. Like for $PM_{2.5}$ described above, the contribution from ships is around 11% of the total number of premature deaths, when based on the DEHM-EVA simulation for 2015. When based on the parallel MATCH-EVA results, the estimated contribution from shipping

is ca. 6%. In the DEHM BAU Baseline, BAU SECA and BAU HFO ban simulations for 2030 this fraction increases somewhat to about 13%, before it decreases to ca. 9% in the 2050 scenarios. In the HiG Baseline scenario the fraction is ca 15% in 2030, with a decrease to 13% in 2050.

The future scenarios discussion here all include the polar diversion as described in Section 3.1.3. To isolate the effect of this,

simulations have been made with the 2050 BAU Baseline scenario with and without the polar diversion. In terms of health effects the additional emission leads to 10-30 premature deaths in the Nordic area.





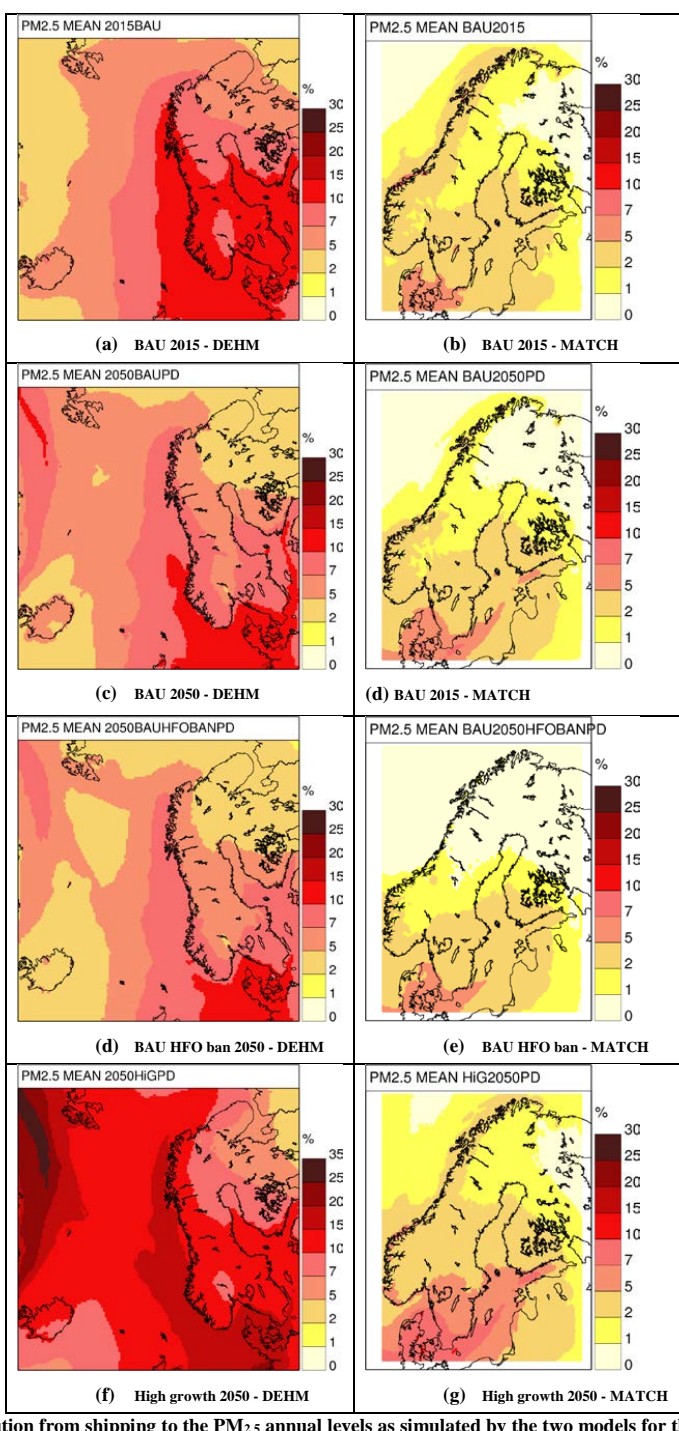

**Figure 11: The contribution from shipping to the PM$_{2.5}$ annual levels as simulated by the two models for the present day and three different shipping emission scenarios for 2050. (Calculated as e.g. (2015BAU-2015BAU_70%/0.3)/2015BAU * 100%, where 2015BAU_70 is a run where the shipping emissions have been reduced to 70%).**





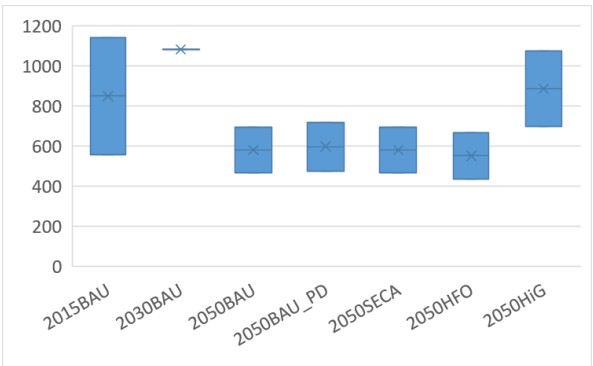

**Figure 12: The estimated number of premature deaths attributable to shipping. The difference between 2050BAU and 2050BAU_PD shows the impact from polar diversion routes. (Note that the 2030BAU and the SECA, HFO and HiG scenarios also includes the polar diversion routes).**

### 3.5 Shipping and related depositions in the Arctic

The second domain in the DEHM model (d02) covers the Arctic region and in the following, the impact of the shipping
scenarios are briefly analysed for two components (BC and total Nitrogen (N) deposition) within the Arctic area, where we here focus on the modelled deposition of these components.

A map of the BC deposition is given in Figure 13a for the DEHM BAU Baseline 2050 simulation (for the d02 domain with a horizontal resolution of 50 km x 50 km). The deposition of BC under both present day and future conditions show the same overall pattern, with a gradient towards the Arctic area, where the deposition level is low (<1 mg/m$^2$). With the included
emissions in the Baseline scenario (including both land-based and shipping emissions), the deposition is projected to decrease by -1 % to -10 % across the Arctic towards 2050. The overall contribution from shipping to the BC deposition in the Arctic is very low and below 1% in most of the Arctic area, as can be seen from Fig 13b (the pattern is very similar for the 2015 simulation). In the simulation including the new diversion shipping routes (Fig 13c), contributions of up to 3% can be seen along the ship routes across the Arctic Sea and in the Baffin Bay, along the coast of Ellesmere Island, Canada and Northwest
Greenland. The contribution will be higher during summer, when the shipping activity along these routes peaks. In 2050 the emission along the new diversion routes will constitute approximately 50% of the BC related to shipping in the area.

As described in Section 3.1 for the Nordic region, the future mitigation scenario with an HFO ban will have the largest impact on the BC emissions and for the Arctic this scenario leads to a decrease of about 12 % compared to the BAU Baseline for 2050 (see Winther et al., 2017 for details).  When comparing the simulations, where the polar diversion route is included, the results
show that an HFO ban will lead to reductions of a few percent in the BC deposition in the Arctic; mainly along the shipping routes (Fig 13d).

In the scenario describing a high growth in the shipping traffic (HiG), the contribution can increase to 5-15 % along the shipping routes as seen in Fig. 13e. Also in areas further away from the routes, a general increase in the contribution from shipping is seen. In an earlier study using a global model and a similar HiG scenario, Browse et al. (2013) found comparable
increases in the contribution from shipping towards 2050 and they conclude that shipping could have a significant impact on the albedo in this area. The deposition of BC on snow and ice decreases the albedo and increases the absorption of incoming solar radiation, which can lead to earlier melting (AMAP, 2011).






**Figure 13: (a) The total BC deposition across the Arctic and Nordic region in the BAU 2050 scenario in the DEHM d02 domain. (b) and (c) displays the contribution from shipping without and with the polar diversion, while (e) and (d) displays the contribution in the HFO ban and High growth traffic scenarios (both including the polar diversion).**


Globally, observations and modelling results indicate that radiative forcing (RF) induced by BC on snow and ice is highest in the mid-latitudes (Bond et al., 2013; Kang et al., 2020). While BC in the atmosphere can lead to a direct RF of +0.71 W m$^2$, its semi- and indirect effects can be up to +0.23 W m$^2$. Sand et al. (2013) showed the Arctic (>60 N) sources of BC leads to a



surface warming of 2.3 K, 1.6 K of which is attributed to the effect from BC deposited on snow and sea-ice. Sand et al. (2016)
estimated that BC in the atmosphere and snow leads to an Arctic warming of 0.48 K.  Both Sand et al. (2013) and (2016)
showed that per unit emissions, sources in the Arctic have a factor of two higher impact on the Arctic warming compared to
sources outside Arctic. These findings imply that increased BC emissions from shipping activities in the Arctic can have a
large effect on the Arctic climate even though these emissions are small compared to global emissions.

Overall, the BC emissions in the Arctic are project to increase towards 2050 in the current study. The SECA scenario will limit
the BC emission by about 3% in the Arctic (North of 60N) in both traffic scenarios, while the HFO ban will have a higher
impact and limit the emissions by up to about 14% in the HiG traffic scenario (Winther et al., 2017). A HFO ban can thereby
limit part of the emissions increase following the HiG traffic scenario, but not all.

The map of total nitrogen (N) deposition is given in Figure 14a for the DEHM BAU Baseline 2050 simulation (the d02 domain,
50 km x 50 km resolution). Like for BC, the deposition of N under both present day and future conditions, show a clear gradient
towards the Arctic area and is generally low (<1 kg N/ha per year) across the Arctic. Within the d02 model domain the changed
emissions between 2015 and 2050 lead to an overall decreasing N deposition across North America/Europe and towards north,
but to an increase in the Russian area. This is reflected in the projected deposition to the Arctic area, where the largest decrease
of about 20% is seen towards the North Atlantic, while a decrease of only a few % is seen for the high Arctic (as the distribution
is very similar in 2015 and 2050, only the latter is shown in Figure 14).

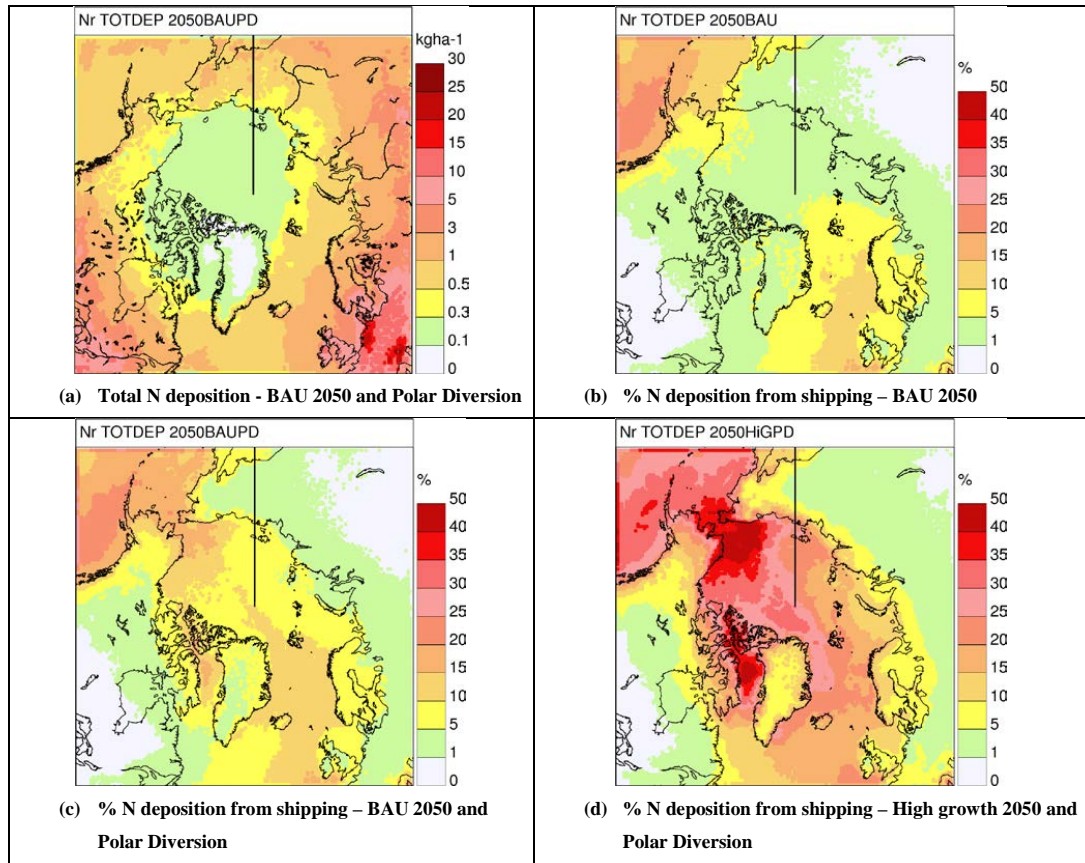

(a)    Total N deposition - BAU 2050 and Polar Diversion

(b)    % N deposition from shipping – BAU 2050

(c)    % N deposition from shipping – BAU 2050 and
Polar Diversion

(d)    % N deposition from shipping – High growth 2050 and
Polar Diversion

**Figure 14: (a) The total Nitrogen deposition across the Arctic and Nordic region in the BAU 2050 Polar diversion scenario in the
DEHM d02 domain. (b) and (c) displays the contribution from shipping without and with the polar diversion, while (d) displays the
contribution in the High growth traffic scenario (including the polar diversion).**




In a previous modelling study with focus on the Canadian Arctic, Gong et al. (2018) estimated the present day annual N deposition to be on the order of 0.2–1 kg N/ha within the Canadian sub-Arctic, and 0.05–0.2 kg N/ha over the Canadian high Arctic, which overall is similar to the levels estimated in the present study.

The shipping scenarios setup in the current study, only impact the $NO_x$ emissions through the development in ship traffic and the new diversion routes. The contribution from shipping in 2050 without (Fig 14b) and with the new routes (Fig 14c), is seen to be slightly higher than for BC and more "widespread" due a lower lifetime of the N-components. The contribution is on the order of 1-10% without the new routes, whereas the contribution reaches up to about 15-20% along the new diversion route in the Baffin Bay, and hence along the coast of Ellesmere Island, Canada and Northwest Greenland. If the traffic develops along

the HiG traffic scenario (Fig 14c), the contribution from shipping increases to more than 20% in large parts of the Arctic and to even above 40% in the Baffin Bay and in part of the Arctic sea. The Arctic ecosystems are adapted to nutrient-poor conditions and critical loads for N depositions are hardly exceeded during present day conditions (Forsius et al., 2010). But as also outlined by Forsius and colleagues, the N cycle in ecosystems are very complex and highly sensitive to increasing N depositions that can lead to significance changes in e.g. inter-species relationships.

**4 Discussion**

Future projections and health assessments are inherently associated with uncertainties. In the following some of the uncertainties and limitations of the presents study are discussed.

The future developments in shipping traffic is in this study based on Corbett et al. (2010), which to our knowledge was the

only dedicated shipping traffic forecast for the Arctic available, when our scenarios were developed. They assume a strong growth of shipping in the Arctic areas. There are, however, several factors, which will have an impact on the popularity of the Arctic ship routes and hence on the future traffic growth. First is the extent of receding sea ice, which makes Arctic routes more manoeuvrable and viable for ships. Second factor involves the geopolitical situation, because local icebreaker assistance may be necessary for safe navigation in Arctic waters. Third are the expenses, fuel, personnel and insurance (Sarrabezoles et

al., 2016) costs, which complicate the evaluations of economic viability of the Arctic sea routes. However, the assumed traffic scenarios of Corbett are used here for consistency with other previous studies, and to study the magnitude of changes concerning the fuel restrictions in the scenarios.

We then combine the shipping scenarios with the ECLIPSE V5a (current legislation) land-based emission scenario to project

future impacts on air pollution levels and human health. However, not only emission changes have an impact on future developments in the air pollution levels. A changed climate will also impact future levels, both directly through altered transport patterns and changed temperature, precipitation and sunshine impacting chemical transformations and removal processes as well as indirectly through changed biogenic emissions and deposition due to changes in growing season of vegetation (Andersson and Engardt, 2010); an important sink of many reactive tropospheric trace gases such as $O_3$. Numerous

previous studies have been conducted to investigate the impact of climate change on air pollution levels in Europe, e.g. Colette et al. (2015) summarizes the climate penalty of $O_3$ in Europe. The studies are conclusive: although climate change until 2050 has an impact on near-surface $O_3$ concentrations (e.g. Langner et al., 2012), particle concentrations (Hedegaard et al., 2013; Lacressoniere et al., 2017), and deposition of N (Simpson et al., 2014), the main factor of future evolution until the mid-century, is overall the emission changes.






There are differences in the results for the current and future scenarios from the two CTMs. MATCH projects a lower exposure in general and the changes to the future differs. For example, MATCH projects a wider spread increase in $PM_{2.5}$ in the Nordic area than DEHM. The fact that we are using two models should be seen as a benefit to illustrate uncertainties in the projections. The main message is robust, but spatial patterns differ and are more uncertain. There may be various reasons for these

differences, ranging from the higher resolution in the MATCH setup, differences in chemical scheme and differences in the long-range transported (LRT) component from continental Europe. The LRT-component is weaker in MATCH, which partly can be due to differences in meteorological forcing, but also due to differences in parameterised deposition processes. It is expected that models with different physical descriptions and setups give some differences in the resulting air pollution distributions and the inclusion of several models hence provides valuable insight on CTM uncertainty. An ensemble of models

usually performs better than individual models (Marecal et al., 2015), and an interval gives an indication of the uncertainties involved. This has been used in other studies, e.g. HTAP/AQMEII (Solazzo et al., 2012a,b; Im et al., 2015a,b) and EDTRENDS (Colette et al., 2017; Vivanco et al., 2018; Ciarelli et al., 2019b) and ENSCLIM (e.g. Simpson et al., 2014; Soares et al., 2015). As illustrated in Figure 11 and 12 the differences between the two models is also clearly seen in the %-contribution from shipping to the overall $PM_{2.5}$ level and the estimated number of premature deaths attributable to shipping. This was also seen

in a recent study focusing on the Baltic Sea region. Here three different CTMs were compared and significant differences were found in the estimated contribution from shipping, which mainly was linked to differences in the schemes for inorganic aerosol formation (Karl et al., 2019).

The health assessment in the current study is based on the modelled air pollution maps with a resolution of 16.67 km x 16.67

km grid across the Nordic area. The estimated number of premature deaths related to air pollution can therefore be somewhat underestimated as the high air pollution levels in urban areas with high population density will not be fully resolved at this resolution. National assessments of health effects for Denmark (see e.g. Ellermann et al., 2020) and the Nordic area (Lehtomaki et al., 2020) made with the same EVA system as applied here, but based on higher resolution air pollution modelling (1 km x 1 km with the Danish UBM model) points towards higher number of premature deaths than in the present study. Additionally,

as described earlier, the EVA system employs a linear exposure-response relationship based on the recommendations of WHO (WHO, 2013). However, recent studies show that the shape of the relative risk vs pollutant concentration can change from region to region, and may not be linear (e.g. Burnett et al., 2018). Some studies have shown that in low pollutant regions, such as the Nordic region, non-linear functions give higher negative impact from air pollution than linear functions, while in high pollutant regions, i.e. China/Africa, linear functions give the highest numbers (Im et al., 2018; Bauer et al., 2019).

Finally, it should also be noted that we in the projected health assessment keep the population data constant (at the 2015 number and distribution), even if previous studies have shown that e.g. an aging population in both the Nordic area and in Europe in general, will lead to a higher sensitivity to air pollution (Geels et al, 2015; Tarín-Carrasco, submitted).

Our focus has been on the Nordic and Arctic region, where we e.g. find increased deposition of N and BC in the Arctic due to

new shipping routes in the future. But these new routes also have the potential to limit the overall $CO_2$ emissions compared to longer routes (Corbett et al., 2010) and air pollution levels and negative health impacts could be lowered in other parts of the world. The study by Sofiev et al. (2018) has e.g. previously shown that especially the Asia population would benefit from ship emission reductions.

The full climatic effects of BC emissions are very complex. As e.g. described in Kühn et al. (2020) BC in the Arctic atmosphere and on snow/ice can lead to a general warming of the climate. But part of this warming can be counteracted by other processes, as aged BC particle e.g. can impact cloud dynamic that have a cooling effect. It has been outside the scope of the current work to make an assessment of the different climate feedbacks related to BC in the Arctic.



**5 Conclusion**

We have setup new global shipping emission scenarios including potential mitigation options either based on additional fuel quality requirements (Heavy fuel Oli ban) or on an expansion of the existing ECA areas. The scenarios are considered in terms of two alternatives for the development in traffic: a Business As Usual (BAU) and a High Growth (HiG) traffic growth scenario and for the years 2015 (our base year), 2020, 2030 and 2050. Overall, the projected increase in traffic is to some degree counteracting the effects of technological developments and fuel requirements leading to different trends for the different

components. When focusing on the Nordic area the projections display that Baseline NOx emissions will go down by approximately 40% (HiG traffic growth) to 60% (BAU traffic growth) towards 2050 due to improved technology. The fuel consumption and $NO_x$ emission totals calculated for the SECA and HFO ban scenario equal the Baseline scenario. For $SO_2$ the Baseline projections includes a decrease on the order of 40-50% in 2050 compared to 2015. Here the largest drop is seen from 2020 due to the global 0.5% Sulphur cap. In all scenario years for $SO_2$, the calculated emissions for the SECA and HFO

ban scenarios are close to 30% lower than the emissions calculated for the Baseline scenario within the Nordic domain. Due to a different development in BC emission factors, the BC emissions increases are 20-70% in the Baseline for 2050 and only a small (1-2%) reduction is obtained in the future years in the SECA scenario in both traffic growth cases. For the HFO ban scenario in 2020*[2030, 2050]* with BAU traffic growth, the BC emissions are 9%*[12%, 16%]* smaller than the Baseline results, while slightly larger emission reductions are obtained in the HiG traffic growth scenario.


       In combination with a scenario for the land-based emissions the two chemistry transport models DEHM and MATCH have been used to simulate the developments in the overall air pollution towards 2050 and also the contribution related to shipping alone. By using the health assessment model EVA, we estimate that for the Nordic area the number of premature deaths related to air pollution will decrease from approximately 9900 (9400-10.400) in 2015 to 7900 (7700-8200) in 2050. The range of the

numbers given in brackets, represents the results for the two models and illustrates the uncertainties related to this kind of assessments e.g. related to differences in the setup of the two CTMs. Changes in climate and population demography are disregarded here, where meteorological and population data for 2015 are used in all simulations.

       Shipping contributes to 1-15% of the $PM_{2.5}$ levels in the Nordic, with highest contribution along the shipping routes and in coastal areas. In terms of health impact, we estimate that shipping emissions leads to about 850 (560-1100) premature deaths

during current day conditions (as a mean over the two models). With the 2050 Baseline BAU scenario this number will decrease to approximately 600 (480-720) cases. Introducing additional mitigation options as assumed in the heavy fuel oil (HFO) ban scenario, the number of premature deaths attributable to shipping emissions will be on the order of 550 (440-670) in 2050. The SECA scenario will have less impact in the Nordic area. If the shipping traffic follows a high growth (HiG) path and no additional mitigation options are introduced, the negative health impact will increase, and the number of premature deaths will

be on the order of 890 (700-1080) in 2050. In terms of health impacts in the Nordic area, the HFO ban BAU traffic scenario can be regarded as the "best case", while the Baseline HiG traffic scenario must be regarded as the "worst case".

       When moving the focus further towards north, we investigate the impacts of new potential ship traffic routes in the future Arctic and the effect of expanding requirements from current SECA areas as well as a HFO ban to the full Arctic region. The

new diversion routes will make the shipping traffic a more important source for BC and N in parts of Arctic, especially along the shipping routes. If the development in traffic follows the high growth path, the simulations show significant increases in the deposition. The mitigation scenarios will limit the contribution from shipping slightly, but the deposition is in general low in the Arctic.

This study addresses the topic, which was recently debated at the International Maritime Organization and reports the results for various emission mitigation options for ships operating in Arctic areas. Health benefits of Arctic HFO ban and an Arctic



SECA have been quantified, considering the health effects in Nordic countries. It is very likely that full benefits of fuel restrictions in Arctic areas will be delayed, because of the recent decision of IMO MEPC75 to relax the fuel requirements for some ships belonging to the Arctic fleets and postpone the full HFO ban until 2029. By setting up these new shipping emissions

scenarios that follows on-going discussions for mitigation options and evaluate the impacts in terms of health impacts in the Nordic region and depositions in the Arctic, this study can add to the science based evaluation of potential mitigation strategies for shipping emissions. It also adds to the recent work of the IMO Fourth GHG study (Faber et al, 2020) and updates regional Arctic emission scenarios and BC emissions from ships, considering the atmospheric transport of pollutants.

**Data availability**

The global shipping emission scenarios have been published at https://zenodo.org/record/4322247#.X9nGi9hKj4I, see Geels et al. (2020).

**Author contribution**

CAG, MW, J-PJ, CA designed the study. MW, JC and J-PJ made the shipping emission scenarios. CAG and CA carried out the simulations and model evaluations with DEHM and MATCH. CAG made the health assessment with contributions from

UI, LMF and JBR. WL prepared the plots. All contributed to the analysis of the results. CAG prepared the paper, with contributions from all co-authors.

**Competing interests**

The authors declare that they have no conflict of interest.

**Acknowledgements**

Nordic Council of Ministers for funding the project Emissions from shiPs and the Impacts on human healTh and envirOnMEnt in the Nordic/Arctic - now and in the future (EPITOME). NordForsk under the Nordic Programme on Health and Welfare. Project 75007: Understanding the link between Air pollution and Distribution of related Health Impacts and Welfare in the Nordic countries (NordicWelfAir). The European Union's Horizon 2020 research and innovation programme under grant agreement No 820655 (EXHAUSTION) and No 874990 (EMERGE). The 2017-2018 Belmont Forum and BiodivERsA joint

call for research proposals, under the BiodivScen ERA-Net COFUND programme, and with the funding organisations AKA (Academy of Finland contract no 326328), ANR (ANR-18-EBI4-0007), BMBF (KFZ: 01LC1810A), FORMAS (contract no:s 2018-02434, 2018-02436, 2018-02437, 2018-02438) and MICINN (through APCIN: PCI2018-093149): granted project BioDiv-Support.

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
