# Peer review of "Projections of shipping emissions and the related impact on air pollution and human health in the Nordic region"

_Atmospheric Chemistry and Physics, 2020_

## Author Comment (AC1)

*Answer to Anonymous Referee #1:*

*First, we want to thank the referee for the positive review and the constructive comments on how to improve the manuscript. In the following, we reply to each comment (the text in italic):*

This MS presents projections of future shipping emissions in different scenarios, as well as their expected impacts on human health. It is very well written and straightforward, with clearly structured objectives. The research topic is relevant and of interest for the scientific community. I may recommend publication, with some suggestions which may help place the authors' results further in context:

*Reply: Great that the referee finds that the paper is well written and of scientific interest.*

- Abstract, "But the question is if this is enough to mitigate the future increase in shipping activities." Please rephrase - the fact that pollutant emissions are decreased in ECAs has no relation with mitigating the increase of shipping activities in the future.

*Reply: We agree and have rephrased that to:*

*"*But the question is if increased shipping in the future will counteract these emission reductions.*"*

- line 27, section 3.4 and conclusions (line 661): to place these numbers (e.g., 850 premature deaths) in context for the reader, please translate them into premature deaths/100000 inhabitants, as done by Fann et al. (2019) and Viana et al. (2020). This would help comparing with other studies and understanding the magnitude of the health impacts reported.

Fann, N., Coffman, E., Hajat, A., Kim, S.-Y., 2019. Change in fine particle-related premature deaths among US population subgroups between 1980 and 2010. Air Qual.Atmos. Heal. 12, 673–682. https://doi.org/10.1007/s11869-019-00686-9;

Viana, M., Rizza, V., Tobías, A., Carr, E., Corbett, J., Sofiev, M., Karanasiou, A., Buonanno, G., & Fann, N. (2020). Estimated health impacts from maritime transport in the Mediterranean region and benefits from the use of cleaner fuels. Environment International, 138, 105670.

*Reply: We agree that this is a nice way to report the numbers. We have added the text below. We also thank for the references given. The Viana et al. paper is very relevant and is now included in the **introduction and Section 3.4.***

Added to line 27: "(corresponding to approximately 37 premature deaths for every 100.000 inhabitants)"

Added to the introduction (line 67): "A health assessment for 8 European Mediterranean coastal cities found that shipping emissions can be related to about 5.5 premature deaths per year for every 100.000 inhabitants in the 8 cities (Viana et al., 2020).

Added to line 477: "A recent study finds that long-term exposure to $PM_{2.5}$ from shipping can be associated with 5.5 premature deaths per 100.000 inhabitants in eight Mediterranean coastal cities per year (Viana et al., 2020). For comparison the 850 deaths in the Nordic area, corresponds to approximately 3.2 premature deaths per year per 100.000 Nordic inhabitants. This is somewhat lower that the number for the Mediterranean cities, but this seems reasonable since several of the include cities in Spain and Italy are located in areas with significant shipping activities."

Added to line 657: "This correspond to a decrease from approximately 37 premature deaths for every 100.000 Nordic inhabitants in 2015 and to approximately 30 in 2050."

Added to line 664: "Seen in relation to the total population in the Nordic this corresponds to about 3 premature deaths per 100.000 inhabitants in 2015 decreasing to about 2 in 2050."

- line 75 (and/or in section 2.1.3): please comment on the likelihood of each of the scenarios proposed.

*Reply: SECA seems politically unlikely because the IMO adopted an HFO ban recently. The SECA, HFO ban and the global sulphur cap aim at reducing sulphur and PM emissions from ships. Introduction of a SECA without the Arctic HFO ban would have allowed the use of Ultra Low Sulphur Fuel Oil (ULSFO) in Arctic waters.*

*HFO ban was agreed at MEPC75 meeting in Nov 2020. This ban includes use and carriage of HFO in Arctic waters from July 1st 2024 onwards. However, there are exemptions to this for countries which have coastlines bordering Arctic waters, until July 2029. The HFO ban will enter in force for all vessels from that date onwards. It should be noted that hybrid fuels, like Very Low Sulphur Fuel Oil (VLSFO) and ULSFO are considered as heavy fuels and cannot be used in Arctic waters once the ban is in force. The use of these fuels has increased considerably since the introduction of global 2020 sulphur cap. With the Arctic HFO ban, vessels operating at high latitudes need to switch to lighter distillate fuels.*

*We have assumed rather marginal growth of LNG vessels in the Arctic, because the lack of refueling infrastructure is a significant challenge. Gas tankers carrying LNG cargoes and using boil-off gas as a propulsion fuel would still be feasible, though. The scenarios developed in this work were aligned with those of the Third IMO GHG study.*

*We have now added some of these considerations to the discussion in Section 4:*

*"*In terms of likelihood, the HFO ban scenario seems most likely at the moment as the IMO has recently (at the MEPC75 meeting in November 2020) adopted a HFO ban. This ban includes use and carriage of HFO in Arctic waters from July 1st 2024 onwards. However, there are exemptions to this for countries with coastlines bordering Arctic waters. The HFO ban will enter in force for all vessels from July 2029. This will, however, require that vessels operating at higher latitudes move away from the use of e.g. hybrid fuels, like Very Low Sulphur Fuel Oil and the use of these fuels has increased considerably since the introduction of global 2020 sulphur cap.*"*

- line 88, "For the Nordic area, we focus mainly on total PM2.5, while for the Arctic we focus on the deposition of nitrogen and black carbon", please justify these choices. Given the health relevance of BC, why was this parameter not included in the analysis for the Nordic area?

*Reply: For the Arctic, we look at the deposition of BC due to the potential climate impact that is widely debated. BC is certainly also interesting in terms of health and in our study it is part of the total PM2.5 used in the health analysis for the Nordic area. To clarify that BC is included in the health study, we have added the following:*

Added line 88: "(the sum of primary emitted components (e.g. black carbon) and secondary formed aerosols)"

- line 89: please clarify the "EVA" acronym.

*Reply: done*

- Sections 2.3 and 2.4: please add at least some quantitative analysis of the performance of the CTMs in the main text, i.e., what is their uncertainty when compared to surface observations? It is good that the details are presented in the Supplement, but a short comment on model validation in the main text would help the reader (e.g., line 242 ", it displays high correlation (insert quantitative data here) with observations while PM concentrations are somewhat underestimated").

*Reply: We agree that this would help the reader. This has been added to both models:*

Added line 226 (the section on DEHM): "(fractional bias for daily values in 2015: -0.09, 0.03 and 0.05 for $PM_{2.5}$, $NO_2$, and $O_3$) and variability (correlations for daily values in 2015: 0.79, 0.73 and 0.93 - see the Supplement for details)"

Added line 256 (the section on MATCH): "(fractional bias of daily mean $O_3$ and correlation coefficients are 6% and 0.80 in Scandinavia in 2015)" and "(for Europe fractional bias and correlation coefficients of daily mean $PM_{10}$, $PM_{2.5}$, $O_3$, $NO_2$ are -11%/0.59, -13%/0.67, 1.6%/0.70, -5.9%/0.58 )."

- line 256, are the terms "acute deaths" and "chronic deaths" the best terms here? Death is usually pretty chronic... maybe "premature deaths due to chronic/acute exposures"?

*Reply: We agree that these commonly applied terms are a bit strange and misleading. We have simply removed the terms acute and chronic deaths. They were not needed as we already define the premature deaths in terms of the exposure: "The number of premature deaths in the system is calculated from short-term exposure to O3, NO2, SO2 and PM2.5 as well as long-term exposure to PM2.5 and NO2."*

- line 303, please explain "For SO2 and BC, the major reason for the emission reductions outside SECA from 2015 to 2050 is the shift from HFO with a Sulphur content of 2.45% in 2015 to HFO with 0.5% Sulphur from 2020 onwards and the consequently reduced emission factors": is the lower S content expected to result in improved combustion efficiency, and therefore lower BC emission factors? Or what is the reason for the decrease in BC?

*Reply: Yes, the less heavy 0.5% Sulphur fuel will due to the composition lead to a more complete combustion. This is important information and we have added the following text*

Added to after line 303: "Compared with the HFO 2.45 % Sulphur fuel, the less heavy 0.5% Sulphur fuel has a smaller amount of heavy organic compounds, hence the fuel combustion becomes more complete and BC emission factors consequently lower."

- line 377 "The total number of premature deaths in the Nordic region in 2015" and Figure 7: please clarify, does this refer to all-cause mortality? Or cause-specific (shipping)? The comparison in Fig 8 suggests it is all-cause.

*Reply: This is due to all the included natural and anthropogenic sources. This has been clarified by adding the text below.*

Added line 375: "This includes all anthropogenic and natural emissions as described in Section 2.1."

Added Fig 7 text: "**(including all main anthropogenic emission sources)**"

- Fig 8: the scale of the Y axis limits the understanding of the actual differences between models. Please find a different type of chart to report this comparison, or discuss the actual numbers in the text.

*Reply: We have now improved the plot (see below), so that is easier to compare the models (and added Y axis label). We keep this kind of plot, as it makes it possible to include the error bars on the EEA estimate, which is important for the comparison. We also include is small discussion of the numbers.*

*Added line 412: "*The DEHM-EVA and MATCH-EVA estimates are both within the error interval of the EEA estimates, but the MATCH-EVA numbers are for all countries lower than EEA and DEHM-EVA. This is in line with the underestimation the MATCH model show for PM in the applied setup.*"*

[Figure]

- Fig 10: image quality should be improved. Please add Y axis lables.

*Reply: Figure 10a has been deleted as the numbers are discussed in the text. Figure 10b has been moved and merged to the former Figure 12 in order to have the information on the contribution from shipping collected in one Figure.*

- Fig 12: image quality should be improved. Please add Y axis lable.

*Reply: Done*

- The section on limitations is very valuable.

*Reply: Thank you. We find it important to discuss the limitations in this kind of study based on projections.*

---

## Author Comment (AC2)

Answer to Anonymous Referee #2:

*First, we want to thank the referee for the positive comments and for concluding that the topic is important and the well written and interesting. We follow the suggestions for revisions that has lead to an improved manuscript. We reply to each comment below (the text in italic):*

GENERAL COMMENTS

The manuscript reports a study that mainly aimed to predict shipping emissions and evaluate their impact on air quality over the Nordic, including the Arctic area, as well as to assess the health impact. The study is very interesting on a very important topic and is well written. Overall, the methodology was well performed. Nevertheless, there are some revisions that I recommend to be performed before publication.

SPECIFIC COMMENTS

1.Introduction

1.1 In the objective there is reference to "…assess the overall impact…". What does this mean? I mortality and morbidity going to be assessed?

*Reply: In the description of the health assessment model, we give details on the health endpoints included. But we agree that more details should be given in the introduction. We have therefore added the text below.*

Added to line 96: "**The focus is here on mortality and the number of premature deaths associated with exposure to air pollution.** "

2. Materials and methods

2.1 The figure with the study area domains should appear in the section that refers to the DEHM model, and no text should be between the section "2. Materials and methods" and the sub-section "2.1 Setup of shipping emissions inventory"

*Reply: We have removed the text and moved the figure as suggested. We have also corrected a mistake in the numbering of the sections 2.2-2.4.*

2.2 Please comment on the likelihood of the future scenarios presented.

*Reply: SECA seems politically unlikely because the IMO adopted an HFO ban recently. The SECA, HFO ban and the global sulphur cap aim at reducing sulphur and PM emissions from ships. Introduction of a SECA without the Arctic HFO ban would have allowed the use of Ultra Low Sulphur Fuel Oil (ULSFO) in Arctic waters.*

*HFO ban was agreed at MEPC75 meeting in Nov 2020. This ban includes use and carriage of HFO in Arctic waters from July 1st 2024 onwards. However, there are exemptions to this for countries which have coastlines bordering Arctic waters, until July 2029. The HFO ban will enter in force for all vessels from that date onwards. It should be noted that hybrid fuels, like Very Low Sulphur Fuel Oil (VLSFO) and ULSFO are considered as heavy fuels and cannot be used in Arctic waters once the ban is in force. The use of these fuels*

*has increased considerably since the introduction of global 2020 sulphur cap. With the Arctic HFO ban, vessels operating at high latitudes need to switch to lighter distillate fuels.*

*We have assumed rather marginal growth of LNG vessels in the Arctic, because the lack of refueling infrastructure is a significant challenge. Gas tankers carrying LNG cargoes and using boil-off gas as a propulsion fuel would still be feasible, though. The scenarios developed in this work were aligned with those of the Third IMO GHG study.*

*We have now added some of these considerations to the discussion in Section 4:*

 *"In terms of likelihood, the HFO ban scenario seems most likely at the moment as the IMO has recently (at the MEPC75 meeting in November 2020) adopted a HFO ban. This ban includes use and carriage of HFO in Arctic waters from July 1st 2024 onwards. However, there are exemptions to this for countries with coastlines bordering Arctic waters. The HFO ban will enter in force for all vessels from July 2029. This will, however, require that vessels operating at higher latitudes move away from the use of e.g. hybrid fuels, like Very Low Sulphur Fuel Oil and the use of these fuels has increased considerably since the introduction of global 2020 sulphur cap."*

3. Results

3.1 There is a very high amount of figures/tables in the main text. Please consider putting some in supplementary material (for example emission factors don´t need to be in the main text; table 2 has the same results as figure5, one should be chosen;.

*Reply: We agree that some of the details given in the tables and figures can be moved to the Supplementary. We have therefore moved Figure 2 (Fuel related emission factors) and Figure 5 (summarizing Table 1 and 2) to a new section on the shipping emissions in the Supplementary. See also below.*

3.2 In figures 7, 8, 10, 12 there are no legends in the axes. Maybe the figures related to health effects could be aggregated in a table where the several scenarios could be more easily compared.

*Reply: Figure 8 has been improved (also following the comments from the other referee). Figure 10a has been deleted as the numbers are discussed in the text. Figure 10b has been moved and merged to the former Figure 12 in order to have the information on the contribution from shipping collected in one figure. We keep this as plots and not a table as we think it is easier to see the difference between the models and the difference between the scenarios in a plot like this.*

3.3 Are the EEA estimates for exposure due to shipping emissions?

*Reply: No, the EEA estimates are for total air pollution – so all emissions sources. We recognizes that that this is not completely clear from the text, so we have added the following text:*

Line 414: From "where premature deaths attributable to $PM_{2.5}$, $NO_2$ and $O_3$ exposure" -> "where premature deaths attributable to *total* $PM_{2.5}$, $NO_2$ and $O_3$ exposure"

Line 419: From "In Figure 8 the EEA estimates for $PM_{2.5}$ mortality in four of the Nordic countries are compared to the current DEHM-EVA and MATCH-EVA estimates based on the Baseline 2015 simulations."-> "In Figure 8 the EEA estimates for $PM_{2.5}$ mortality in four of the Nordic countries are compared to the

current DEHM-EVA and MATCH-EVA estimates based on the Baseline 2015 simulations *including both land based and shipping emissions*."

4. Discussion

4.1 This section should include a subsection with the uncertainties related to all the estimations performed.

*Reply: We agree that a discussion of the uncertainties related to this kind of assessment study, is very important.  In Section 4 we already state  "*Future projections and health assessments are inherently associated with uncertainties. In the following some of the uncertainties and limitations of the presents study are discussed." *So we already discuss the uncertainties related to the different elements of our study. In Section 3.2 we also give the uncertainties given by EEA. In our conclusion we also state "*By using the health assessment model EVA, we estimate that for the Nordic area the number of premature deaths related to air pollution will decrease from approximately 9900 (9400-10.400) in 2015 to 7900 (7700-8200) in 2050.*"  And "*The range of the numbers given in brackets, represents the results for the two models and illustrates the uncertainties related to this kind of assessments e.g. related to differences in the setup of the two CTMs.*"